# FeCu dual-single-atom catalyst promotes gradient H$_2$O$_2$ activation for enhanced methane oxidation to methanol

Haonan Zhang[1,4], Shuai Wang[2,4], Yang Li[1], Hongjie Qin[1], Mingwang Wang[1], Qinghai Chen[1], Boshi Zheng[1], Shuxu Zhu[1], Pengye Zhang[1], Chaoqun Gu[1], Yunyun Li[1], Qi Hua[1], Mingbo Wu ®[1,3] ✉ & Wenting Wu ®[1] ✉

Hydrogen peroxide is an attractive and sustainable oxidant, yet its effective application in inert alkane oxidation is limited by the inability to precisely match the distribution, concentration, and reactivity of generated oxygen species with substrate activation requirements. Herein, a dual single-atom catalyst, FeCu/ZSM-CI, in which atomically dispersed Fe and Cu are spatially separated within the microporous framework of ZSM-5, with Fe located in the inner channels and Cu on the external surface, thereby enabling a controlled H$_2$O$_2$ activation gradient. This spatial configuration induces differentiated reactive oxygen species evolution: high-valent Fe=O and •OOH species form in the interior to activate methane into CH$_3$OOH, while surface Cu sites selectively convert CH$_3$OOH into methanol, mitigating overoxidation pathways. The optimized FeCu/ZSM-CI catalyst achieves a methanol yield of 20.2 mmol g$_{cat}^{-1}$ h$^{-1}$ with 90.1% selectivity and a remarkable H$_2$O$_2$ utilization efficiency of 74.6%. Mechanistic studies combining kinetic isotope effects, scavenger assays, in-situ EPR/DRIFTS, and DFT calculations reveal that Fe-Cu synergy shifts the rate-determining step from H$_2$O$_2$ activation to C-H bond activation. These findings establish a generalizable strategy for manipulating ROS spatial distribution via spatial-configuration-driven synergy and a transferable design principle, offering new insights for designing advanced catalysts for selective hydrocarbon oxidation under ambient conditions.

Hydrogen peroxide (H$_2$O$_2$) is a commonly used oxidant in chemical synthesis and is regarded as one of the top 100 most important chemicals globally due to its effectiveness and eco-friendliness[1–4]. However, improving both selectivity and rate simultaneously can be challenging, particularly for selectively oxidizing inert alkane[5–7]. Excessive use of H$_2$O$_2$ is often employed to enhance the conversion rate, but this reduces the selectivity and efficiency of H$_2$O$_2$ utilization[8–11]. This rises from two main issues: first, single reactive

oxygen species (ROS) cannot effectively activate and selectively convert C-H bonds of inert alkane independently, despite many excellent works have been devoted into the selective conversion of H$_2$O$_2$ into specific ROS, such as superoxide radicals (•O$_2^-$/•OOH), hydroxyl radicals (•OH) and high valent metal-oxygen species (M = O)[9,12,13]. For example, •OH and high-valent metal-oxygen sites have a strong ability to activate C-H bonds in alkanes, and have the ability to selective conversion C-H bond at low concentration[14], but they tend to over-

[1]State Key Laboratory of Heavy Oil Processing, College of Chemistry and Chemical Engineering, Institute of New Energy, China University of Petroleum (East China), Qingdao, P. R. China. [2]China Energy Engineering Group Jiangsu Power Design Institute Co.Ltd, Jiangsu, P. R. China. [3]State Key Laboratory of Advanced Optical Polymer and Manufacturing Technology, College of Chemical Engineering, Qingdao University of Science & Technology, Qingdao, P. R. China. [4]These authors contributed equally: Haonan Zhang, Shuai Wang. ✉e-mail: wumb@upc.edu.cn; wuwt@upc.edu.cn

oxidize into acid or $CO_2$ if their concentration increases for enhancing the reaction rate[15–17]. This leads to the second issue that high concentrations of $H_2O_2$ or ROS do not always match those of reactants[18]. To date, almost no strategy has been proposed to systematically solve the matching problem of the distribution, concentration, and kinds of ROS with reactants.

The fast expansion of renewable energy facilitates a shift from natural gas, a conventional energy source, to chemical feedstocks[19–21]. Since methane (primary component of natural gas) is usually located in distant and dispersed areas, it is better to directly oxidize it into methanol with low boiling point and easy separation for convenient transportation and storage, rather than using the conventional method of reforming it into syngas at high temperatures and then Fischer-Tropsch synthesis[22–24]. Molecular sieves are crucial catalysts (e.g., methane conversion) due to their diverse pore structures, which provide distinct reaction sites and ways to regulate reactant concentration[25–27]. ZSM-5, for example, possesses both inner small and external large pore structures. According to Fick's law, $H_2O_2$ diffuses more in external surface with higher concentrations and less in inner pores with lower concentrations[28,29]. Therefore, by placing functional metal sites more precisely within the pores, ROS concentration can be controlled through diffusion rather than only adjusting $H_2O_2$ amounts.

What metal sites can efficiently promote $H_2O_2$ evolution for activating C-H bonds and selectively converting methane? Non-noble metal iron (Fe) nanoparticles or clusters incorporated within the nanopores of ZSM-5 can react with $H_2O_2$ and are prone to generate •OH for the activation of C-H bond[30]. Reducing Fe site size to binuclear Fe or individual Fe atom could promote the decomposition of $H_2O_2$ from •OH to high valent Fe-O, significantly enhancing methane activation[31,32]. This may be due to the longer intrinsic lifetime of high valent Fe-O compared to •OH (100 ns), providing more reaction time to activate C-H bond[23]. In addition, extra-framework Fe sites in ZSM-5 pores via a template-free synthesis strategy show over 5 times higher methane conversion rate than that Fe sites in the framework of ZSM-5, which provide good reference for precise fabrication of Fe sites to activate C-H bond into •$CH_3$[33].

At this juncture, a new challenge arises: how to control $H_2O_2$ evolution to improve product selectivity, particularly in the conversion of methane into $CH_3OH$? Gradient $H_2O_2$ evolution into mild ROS (e.g. •OOH) is preferable for selectively converting methane into $CH_3OOH$ instead of risking over-oxidation with •OH that reacts with •$CH_3$ and easily form $CH_3OH$, $HCOOH$ and $CO_2$[34,35]. To enhance the product selectivity, it is better to separate the distribution of •OOH and •OH. This allows for $CH_3OOH$ to diffuse into a milder condition for subsequent conversion into $CH_3OH$. Hutchings and Yu et al. found that Cu ions impregnated into molecular sieve (e.g. Fe/ZSM-5) can mildly and selectively convert $H_2O_2$ into •OH, significantly increasing methanol yield up to 85%, providing a good suggestion for selective conversion of $CH_3OOH$[30,36]. However, the efficiency of $H_2O_2$ conversion is only 3.39%, and the significant residual amounts of $H_2O_2$ easily cause over oxidation. Therefore, there is much room left for improvement in both $CH_3OH$ selectivity and $H_2O_2$ utilization efficiency[37].

By optimizing the combination of Fe active sites anchored in the extra framework of inner small pores using the crystal seed method and Cu anchored in the external large pores of ZSM-5 via impregnation methods, $CH_3OH$ yield reaches 20.2 mmol $g_{cat}^{-1}$ $h^{-1}$ with a selectivity of 90.1% in direct oxidation of methane, representing a breakthrough in simultaneously enhancing selectivity and reaction rate. More importantly, the utilization efficiency of $H_2O_2$ reached 74.6%, much higher than most reported results. Experiments and theoretical calculations show that Fe is more reactive with $H_2O_2$, forming high-valent Fe-O and •OOH species that generate $CH_3OOH$ for efficient $H_2O_2$ utilization in inner small pores. Cu has lower reactivity but selectively converts $CH_3OOH$ into $CH_3OH$, promoting selective conversion to methanol in external larger pores at higher $H_2O_2$ concentrations. Combining $H_2O_2$

diffusion and functional metal site distribution can improve gradient $H_2O_2$ evolution to increase both methane conversion rate and selectivity, which provide a new perspective to enhance the selective conversion and efficient utilization of $H_2O_2$.

## Results

### Catalytic performance in direct and selective methane oxidation

To achieve gradient $H_2O_2$ decomposition and improve utilization efficiency, we confined Fe species inside the porous channels of ZSM-5 via an in-situ seed-assisted synthesis method (Noted: C), while loading Cu species on the external surface through impregnation (Noted: I), constructing a spatially segregated FeCu/ZSM-CI catalyst. For comparison, a series of FeCu catalysts were prepared using different synthetic approaches (see Methods). Methane directly oxidation to methanol (DOM) was employed as a model reaction to investigate the gradient $H_2O_2$ activation mechanism.

To evaluate the catalytic performance of FeCu/ZSM-CI, direct oxidation of methane was conducted using $H_2O_2$ as the oxidant in a 50 mL autoclave reactor at 80 °C. Control experiments confirmed that no reaction occurred with alternative oxidants (e.g., $O_2$) or in the absence of $H_2O_2$, $CH_4$, or the catalyst (Supplementary Table 1), demonstrating that both $H_2O_2$ and the catalyst are indispensable for methane conversion. After optimizing key parameters (Supplementary Fig. 1–3), including $H_2O_2$ concentration, reaction temperature, methane pressure, catalyst dosage, and time, the best performance was achieved under the following conditions: 20 mL of 0.1 M $H_2O_2$, 5 mg of catalyst, 80 °C, 3.5 MPa $CH_4$ pressure, and a 3 h reaction time. Under these conditions, FeCu/ZSM-CI exhibited exceptional activity, producing $CH_3OH$ with a yield of 20.2 mmol $g_{cat}^{-1}$ $h^{-1}$ and 90.1% selectivity (Fig. 1a), which was further confirmed by $^{13}C$ NMR (Supplementary Fig. 3b). The methane conversion rate is 1.2% (Supplementary Fig. 4) and the overall methanol yield superior to many state-of-the-art systems (Supplementary Table 2).

To study the influence of metal type on the catalysis performance, monometallic decorated H-ZSM-5 catalysts were employed under the optimal reaction conditions mentioned above (Fig. 1a). When catalyzed by Fe/ZSM-I (Fe primarily located on the external surface of ZSM-5), the yield of total $C_1$ oxidation was 12.1 mmol $g_{cat}^{-1}$ $h^{-1}$, higher than those of Cu/ZSM-I (2.2 mmol $g_{cat}^{-1}$ $h^{-1}$) and pure H-ZSM-5 (0.6 mmol $g_{cat}^{-1}$ $h^{-1}$). Similar results could also be observed in Fe/ZSM-C (16.1 mmol $g_{cat}^{-1}$ $h^{-1}$) and Cu/ZSM-C (1.9 mmol $g_{cat}^{-1}$ $h^{-1}$). This suggests that the metal active site plays a crucial role in $CH_4$ conversion, and Fe is more effective than Cu regardless of whether the catalyst was prepared through impregnation or crystal seed method. However, when catalyzed by Fe/ZSM-I, HCOOH was the main liquid product, and the selectivity of $CH_3OH$ was only 18.4%, which is much lower than that of Cu/ZSM-I (98.6%). When the Fe site was changed from external surface to inner pores in Fe/ZSM-C, the selectivity of $CH_3OH$ increased to 41.6%, but it still remained lower than that of Cu/ZSM-C (96%). It indicates that Cu could maintained high methanol selectivity, and metal location could also influence methanol selectivity. Therefore, combining both Fe and Cu may improve both the yield and selectivity of methanol.

Further studies were conducted to investigate the influence of Fe and Cu location on catalytic performance. FeCu/ZSM-CC (simultaneous in-situ growth of Fe and Cu metals) achieved 89.1% selectivity for $CH_3OH$, but the yield was only 10.4 mmol $g_{cat}^{-1}$ $h^{-1}$ (Fig. 1a). This low yield may be due to competitive adsorption of $CH_4$ and $H_2O_2$ at Fe and Cu sites within a confined space. The unsatisfactory $CH_3OH$ yield (9.6 mmol $g_{cat}^{-1}$ $h^{-1}$ and 77.1%) observed for FeCu/ZSM-II (simultaneous impregnation of metals Fe and Cu) may also be attributed to this reason. Additionally, FeCu/ZSM-II has lower yield and selectivity for $CH_3OH$ than FeCu/ZSM-CC, suggesting different locations in H-ZSM-5 may have distinct reaction processes. As Fe promotes methanol conversion and Cu enhances methanol selectivity, the presence of both Fe

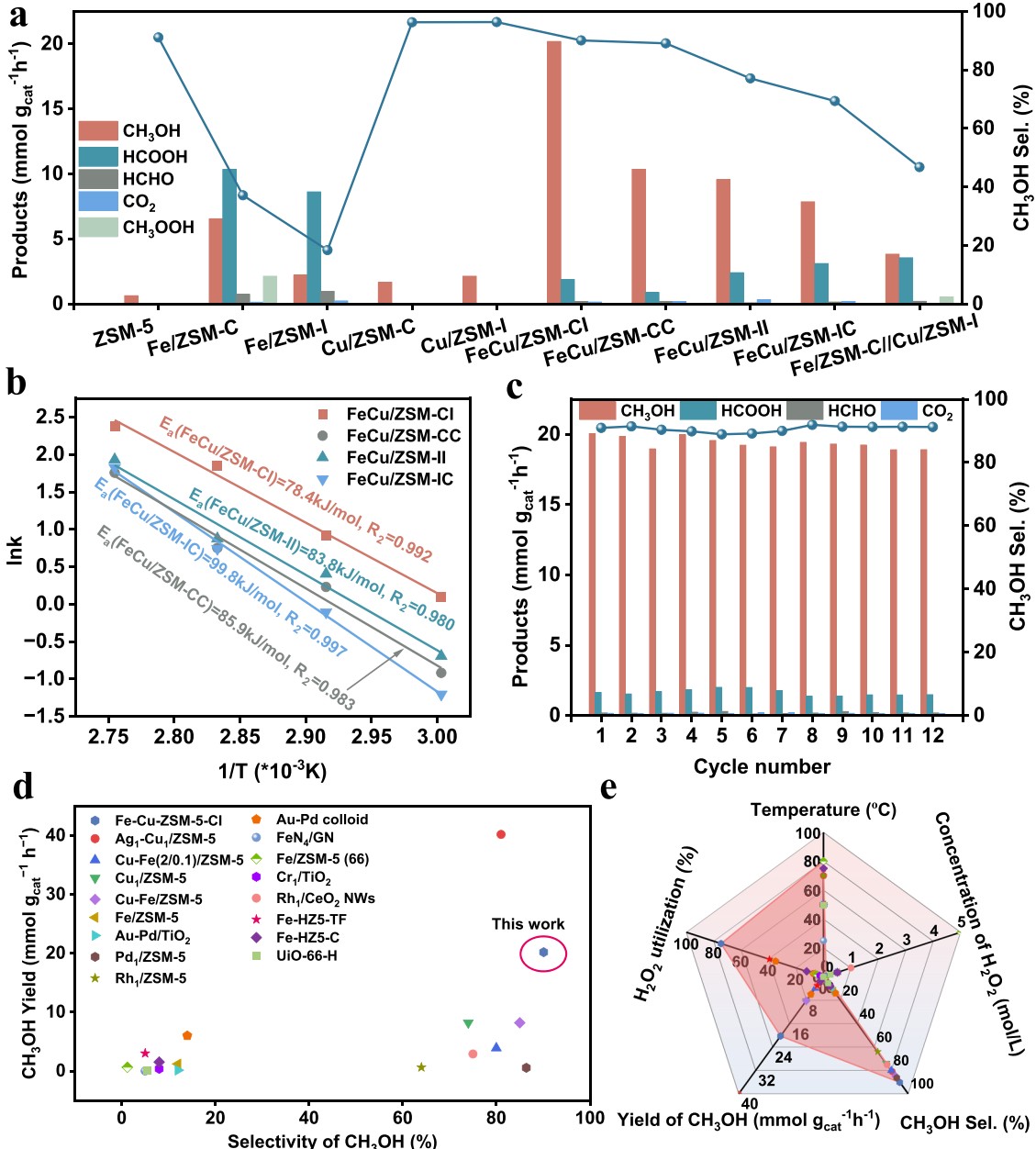

**Fig. 1 | Catalytic performances for direct oxidation of methane. a** Liquid product yields and methanol selectivity on different catalysts, Fe/ZSMC//Cu/ZSM-I represent the physical mixture of Fe/ZSMC and Cu/ZSM-I catalysts. **b** Arrhenius plots for $CH_4$ oxidation over FeCu/ZSM-CI, FeCu/ZSM-CC, FeCu/ZSM-II, and FeCu/ZSM-IC. **c** Cyclic experiment. Reaction Condition: 5 mg catalysts dispersed in 20 mL of 0.1 mol $L^{-1}$ $H_2O_2$ aqueous solution, 80 °C, and 3.5 MPa $CH_4$. **d, e** Comparisons with the representative catalytic performances for methanol yield and selectivity. Numbers in square brackets correspond to the entry numbers in Supplementary Table 3.

and Cu in different locations may increase the yield and selectivity of methanol simultaneously.

The yield and selectivity of $CH_3OH$ for FeCu/ZSM-IC (in-situ growth of Cu and then impregnation anchoring of Fe) was only 69.3% and 7.9 mmol $g_{cat}^{-1}$ $h^{-1}$. In contrast, FeCu/ZSM-CI (in-situ growth of Fe and impregnation anchoring of Cu) exhibited significant improvement with a much higher yield and selectivity of $CH_3OH$ at 90.1% and 20.2 mmol $g_{cat}^{-1}$ $h^{-1}$, surpassing other catalysts. Further investigation was conducted by physically mixing Fe/ZSM-C and Cu/ZSM-I to explore the interaction between Fe and Cu. The results showed a methanol yield of only 3.8 mmol $g_{cat}^{-1}$ $h^{-1}$ with a selectivity of 46.8%, which is significantly lower than that of FeCu/ZSM-CI (Fig. 1a). This further demonstrates the important role of the synergistic effect between Fe and Cu in enhancing both methanol yield and selectivity.

Generally, $H_2O_2$ and $CH_4$ follow Fick's diffusion law in ZSM-5 pores, with lower concentration in the inner pores than the external surface. Therefore, Fe with higher methanol conversion ability was placed in inner pores with lower concentrations of $CH_4$ and $H_2O_2$, while Cu with lower methanol conversion ability but higher $CH_3OH$ selectivity was placed on external surface with higher concentrations of $CH_4$ and $H_2O_2$. This arrangement facilitates gentle and effective utilization of $H_2O_2$ while preventing overoxidation. The utilization efficiency of $H_2O_2$ is estimated to reach 74.6%. The self-decomposition of $H_2O_2$ is a major factor limiting its oxidizing efficiency. The Fe/ZSM-C catalyst generated the highest $O_2$ yield (≈ 0.28 mmol), whereas the Cu/ZSM-I catalyst produced only 0.08 mmol of $O_2$, indicating its lower catalytic activity toward $H_2O_2$ decomposition. Notably, the FeCu/ZSM-CI catalyst yielded 0.32 mmol of $O_2$. Compared to Fe/ZSM-C (Supplementary Fig. 5),

this suggests that the incorporation of Cu mitigates $H_2O_2$ self-decomposition, thereby improving its effective utilization in oxidation.

To ensure a fair comparison of methanol selectivity among the FeCu/ZSM-CI, FeCu/ZSM-CC, FeCu/ZSM-II, and FeCu/ZSM-IC samples, evaluations were conducted at an equivalent activity level (-25-27 mmol $g_{cat}^{-1}$ $h^{-1}$, Supplementary Fig. 6). FeCu/ZSM-CI exhibited outstanding methanol selectivity, exceeding 90%, which is significantly higher than that of the other catalysts. This effectively demonstrates the advantage of the configuration where Fe species are located inside the zeolite framework and Cu species are distributed on the external surface in enhancing methanol selectivity. Subsequently, kinetic measurements were performed to investigate the origin of the activity differences (Fig. 1b). FeCu/ZSM-CI exhibited the lowest apparent activation energy ($E_a$) of 78.4 kJ mol$^{-1}$, followed by FeCu/ZSM-II (83.8 kJ mol$^{-1}$), FeCu/ZSM-CC (85.9 kJ mol$^{-1}$), and FeCu/ZSM-IC (99.8 kJ mol$^{-1}$). These results indicate that precise spatial positioning of Fe and Cu dual atoms not only ensures high catalytic activity but also significantly enhances methanol selectivity.

To verify the heterogeneous nature and stability of the catalyst, a series of control experiments were conducted. In the absence of catalyst, as well as in the presence of iron acetylacetonate, FeCl₂, Cu(NO₃)₂, or their mixtures, no catalytic activity was observed (Supplementary Table 4). Hot filtration tests further confirmed the heterogeneous nature of the catalytic system, as no additional product formation was detected after 1 h of reaction following catalyst removal or ZSM-5 addition (Supplementary Fig. 7). Moreover, FeCu/ZSM-CI demonstrated excellent recyclability, with no significant loss in CH₃OH yield or selectivity over twelve consecutive reaction cycles (Fig. 1c). Inductively coupled plasma (ICP) analysis revealed negligible leaching of Fe and Cu species after the reaction (Supplementary Table 5), further confirming the structural robustness of the catalyst. Overall, FeCu/ZSM-CI exhibits high stability under reaction conditions for methanol production, and its catalytic performance surpasses that of most previously reported catalysts (Fig. 1d, e).

The high cost of $H_2O_2$ limits the practical industrial application of direct methane-to-methanol systems using $H_2O_2$ as the oxidant. Consequently, methane oxidation using $O_2$ as an oxidant is of significant interest (Supplementary Fig. 8). Under reaction conditions of 210 °C with $O_2$ as the oxidant, methane was successfully oxidized to methanol and acetic acid, thereby avoiding the use of $H_2O_2$. Notably, the product distribution strongly depends on the catalyst type. Fe and Cu monometallic catalysts primarily produce methanol, whereas the bimetallic FeCu catalyst exhibits high selectivity toward acetic acid, with a yield of 19 µmol $g_{cat}^{-1}$ and selectivity of 84.6%. These results confirm that the spatially distributed bimetallic FeCu/ZSM-CI catalyst exhibits excellent adaptability, performing effectively not only in $H_2O_2$ based systems but also in $O_2$ driven oxidation systems through reaction condition optimization.

## Characterization of FeCu/ZSM-CI catalyst

To elucidate the structural properties of the as-prepared catalysts, comprehensive characterizations were conducted. The XRD pattern of FeCu/ZSM-CI (Supplementary Fig. 9) retains characteristic ZSM-5 framework peaks, confirming high dispersion of Fe and Cu species within the zeolite matrix. Consistent with this, HRTEM and SEM-EDS mapping (Supplementary Figs. 10–13) revealed no detectable nanoparticles or clusters, indicating atomic-scale anchoring of metals on H-ZSM-5. Strikingly, atomic-resolution AC-HAADF-STEM imaging of Fe/ZSM-C, Cu/ZSM-I, and FeCu/ZSM-CI (Fig. 2a–c) unveiled a distinct spatial arrangement: Fe species predominantly reside within internal zeolite channels, while Cu species enrich the external surface.

To quantify this distribution, semi-quantitative SEM-EDS mapping measured surface Fe and Cu contents at 0.08 wt% and 0.49 wt%, respectively. In contrast, bulk ICP-AES analysis yielded 0.67 wt% Fe and 0.50 wt% Cu (Supplementary Table 5). The significant disparity in Fe content (surface vs. bulk) strongly supports the encapsulation of Fe within internal channels, while the consistent Cu concentration indicates its predominant surface anchoring (Fig. 2d–f).

As the electronic properties of active sites are determined by their structures and coordination environments, X-ray photoelectron spectroscopy (XPS) and X-ray absorption fine structure (XAFS) spectroscopy were applied for further investigation (Supplementary Figs. 14–16). XPS revealed that Fe is present in the +3 oxidation state, as indicated by peaks at 711.3 eV and 724.5 eV corresponding to Fe2$p_{3/2}$ and Fe2$p_{1/2}$, respectively[38,39] (Supplementary Fig. 14). The Cu valence state was found to be between +1 and +2 based on a peak centered at 933.5 eV in the Cu2$p_{3/2}$ spectra (Supplementary Fig. 15), which is higher than that of Cu$^+$ but lower than Cu$^{2+}$[40,41].

For further exploring the coordination environment of Fe and Cu species, XAFS analysis was performed (Supplementary Fig. 16a), Fe K-edge of X-ray Absorption Near Edge Structure (XANES) showed that the Fe absorption edge at 7115 eV is similar to that of $Fe_2O_3$, indicating the valent state of Fe is +3. At the same time, the peak corresponding to the characteristic leading edge of the Fe element at 7114 eV attributed to $Fe_2O_3$ is not obvious, indicating that its structure does not conform to that of a binuclear iron tetrahedron of $Fe_2O_3$[42]. Simultaneously, it was demonstrated that the Fe sites did not substitute for aluminum during the synthesis process, thereby preserving the integrity of the molecular sieve framework; instead, they were situated externally to the structure. The coordination environment was analyzed using EXAFS curve fitting techniques, which revealed two prominent peaks relating to Fe-O scattering at distances of 1.6 Å and 2.6 Å (Supplementary Fig. 17) with coordination numbers of 3.1 and 2.8 respectively (Supplementary Table 6); no peaks associated with Fe-Fe or Fe-O-Fe scattering were detected (Supplementary Fig. 16b). The Fourier Transform (FT) K³-weighted extended EXAFS spectrum for iron was analyzed using the Fe-O backscattering path (Supplementary Fig. 17). The optimal fitting results indicate that the Fe-O bond corresponding to a coordination number of 3.1 at a distance of 1.6 Å pertains to Fe-O-H, which corresponds to the peaks of 715.7 eV and 728.0 eV attributed to Fe(OH)$_x$ (where x = 1, 2, or 3) shown in the XPS spectra (Supplementary Fig. 14)[38]. While the bond corresponding to a coordination number of 2.8 at a distance of 2.6 Å is attributed to Fe-O-Al. It can be seen that the Fe within the FeCu/ZSM-CI catalyst is classified as extra-framework mononuclear iron, exhibiting a total coordination number of 6. A combination of iDPC-STEM, SEM-EDS mapping, and ICP-AES confirmed that Fe exists as single atoms within the inner pores of zeolite.

Similarly, the XANES spectrum of the Cu element reveals that the absorption edge for Cu (Supplementary Fig. 16c) is situated between that of $Cu_2O$ and CuO. Notably, there is no discernible peak at 8977 eV, indicating that the copper present in FeCu/ZSM-CI does not exhibit a high oxidation state. This observation aligns with the XPS data phase revealing Cu$^{\delta+}$ (1 <δ <2) (Supplementary Fig. 15). Additionally, in the EXAFS spectrum (Supplementary Fig. 16d) corresponding to the Cu site, a characteristic peak related to the Cu-O bond can be identified, with its value measured at 1.57 Å. Upon fitting the Cu-O backscattering path, it was determined that the coordination number of cupric ions is 3 (Supplementary Fig. 17, Supplementary Table 6). The presence of Fe and Cu as isolated single atoms was further confirmed by wavelet transform (WT) analysis (Supplementary Fig. 18–19).

Additionally, a series of characterizations (XRD, UV, IR and XANES) of the reaction after reaction (Supplementary Figs. 20–26), reveals that the Fe and Cu sites within the catalyst are still anchored within molecular sieve channels after cycling experiments since there were no observed changes in their structure or morphology. Therefore, the FeCu/ZSM-CI catalyst achieves precise spatial control of atomically dispersed Fe (inside) and Cu (outside) sites through a combined seed-impregnation synthesis. The resulting dual monatomic

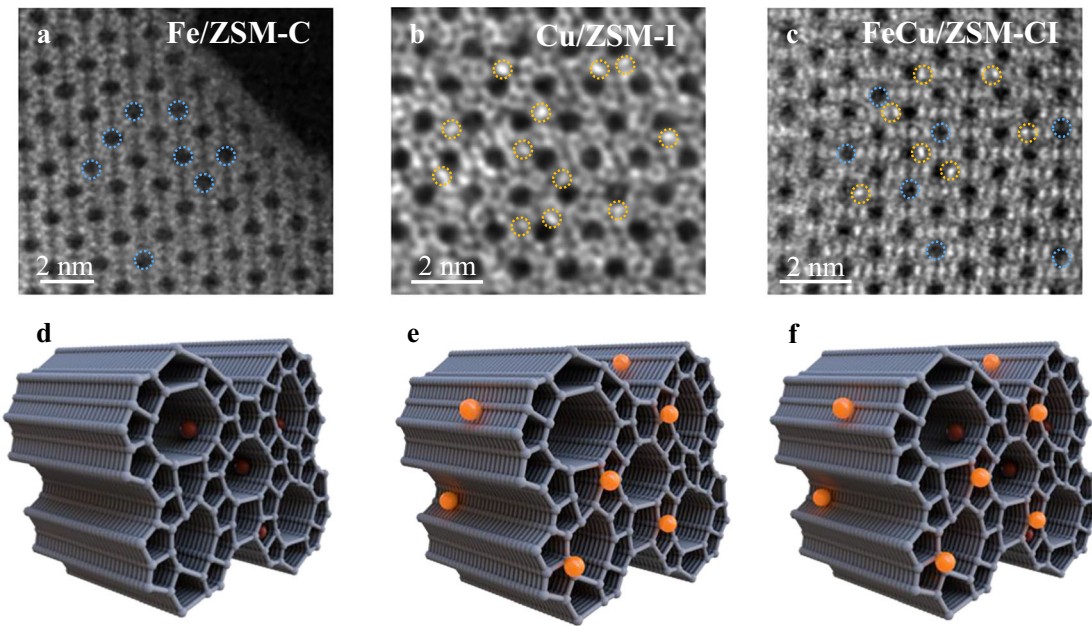

**Fig. 2 | Catalyst model. a–c** iDPC-STEM images of Fe/ZSM-C, Cu/ZSM-I, and FeCu/ZSM-CI. **d–f** Models of Fe/ZSM-C, Cu/ZSM-I, and FeCu/ZSM-CI catalysts.

structure, characterized by its distinct spatial metal sites, facilitates efficient utilization of the $H_2O_2$ gradient while optimizing the reaction pathway for methane. Ultimately, this leads to enhanced selectivity and yield in converting methane to methanol.

## Catalytic mechanism
### Identification of the active species
During $H_2O_2$ activation, the metal site can induce the homolytic or heterolytic O-O bond cleavage, generating diverse reactive species, including •OH, •OOH radicals, and high-valent metal-oxo intermediates. To identify the catalytically relevant species in the FeCu/ZSM-CI/$H_2O_2$ system, we combined electron paramagnetic resonance (EPR) spectroscopy, scavenger experiments, electrochemical analysis, and quasi-in situ characterization.

EPR spectroscopy was first employed to probe radical species formed during $H_2O_2$ activation. As shown in Fig. 3a, when 2,2-dimethyl-1-oxido-3,4-dihydropyrrol-1-ium (DMPO) was used as the trapping agent, only the signals of DMPO-OH and DMPO-•$O_2^-$ adducts were observed in the aqueous solution. In the absence of $H_2O_2$, no •OH or •$CH_3$ were observed under any conditions tested (Supplementary Fig. 27), confirming that methane activation does not occur spontaneously. When $H_2O_2$ was added without the catalyst (Fig. 3b), only •OH signals were detected, and subsequent methane addition did not generate •$CH_3$ radicals. In contrast, upon introducing FeCu/ZSM-CI, clear •$CH_3$ signals emerged alongside •OH, indicating that methane activation proceeds through a catalyst-mediated pathway rather than direct oxidation by free hydroxyl radicals. This behavior suggests that cooperative Fe-Cu interactions enable C-H bond cleavage via activated oxygen species beyond simple •OH chemistry.

The relative importance of individual ROS was further evaluated through scavenger experiments (Fig. 3c). Salicylic acid, p-benzoquinone, and DMSO were employed to quench •OH, •$O_2^-$, and high-valent metal-oxo species, respectively. Notably, DMSO addition caused a pronounced drop in methanol productivity over FeCu/ZSM-CI, from 22.40 to 7.40 mmol $g_{cat}^{-1}$ $h^{-1}$, highlighting the dominant role of high-valent metal-oxo species in methane activation. Scavenging •OH and •$O_2^-$ also led to measurable decreases in liquid-phase products, indicating that multiple ROS participate cooperatively in the reaction network. Since •OH can oxidize dimethyl sulfoxide (DMSO) to DMSO-

OH, while high-valent metals (Fe=O) can oxidize DMSO to $DMSO_2$ (Supplementary Fig. 28)[43]. In the DMSO oxidation system (Fig. 3d), both Fe/ZSM-C and FeCu/ZSM-CI catalysts are capable of oxidizing DMSO to $DMSO_2$, whereas the $H_2O_2$ and the Cu/ZSM-I catalyst alone cannot. This indicates that Fe is oxidized to a high-valent Fe=O species during the process. Complementary in-situ cyclic voltammetry revealed distinct redox features at 0.5-0.7 V for Fe/ZSM-C and FeCu/ZSM-CI upon $H_2O_2$ addition (Fig. 3e, Supplementary Fig. 29), which were absent for Cu/ZSM-I and pristine ZSM-5, directly evidencing dynamic Fe valence cycling under reaction conditions. Consistently, in-situ EPR showed significantly stronger •$CH_3$ signals over Fe/ZSM-C than over Cu/ZSM-I (Supplementary Fig. 30), confirming that Fe-based centers serve as the primary sites for C-H bond activation.

To directly monitor iron oxidation-state evolution during catalysis, quasi-in situ high-frequency EPR (HF-EPR, 240 GHz, 15 K) measurements were performed (Supplementary Fig. 31). The fresh Fe/ZSM-C catalyst exhibited a characteristic signal attributable to isolated high-spin $Fe^{3+}$ (S = 5/2), arising from its $d^5$ electronic configuration in a distorted ligand field. Upon $H_2O_2$ treatment, this signal decreased markedly and eventually disappeared, indicating conversion of EPR-active $Fe^{3+}$ into an EPR-silent species. Although $H_2O_2$ could in principle reduce $Fe^{3+}$ to $Fe^{2+}$, the combined DMSO oxidation and in-situ CV results support oxidation of $Fe^{3+}$ to a high-valent $Fe^{(IV)}$ = O species with an integer spin state (S = 2), which is intrinsically difficult to detect by conventional EPR due to large zero-field splitting.

Subsequent introduction of methane led to recovery of the $Fe^{3+}$ signal, demonstrating that the high-valent $Fe^{(IV)}$ = O species functions as an active oxygen carrier that abstracts hydrogen from $CH_4$ and is concomitantly reduced back to $Fe^{3+}$. HF-EPR thus directly captures the reversible valence cycle **$Fe^{3+}$ → (EPR-silent) $Fe^{(IV)}$ = O → $Fe^{3+}$**, providing compelling spectroscopic support for a radical abstraction mechanism involving a transient Fe-O-$CH_4$ intermediate.

Overall, these results demonstrate that $H_2O_2$ activation within the zeolite framework generates both high-valent metal-oxo species and oxygen-centered radicals, whose identities and reactivities are dictated by the metal centers. Precise control over these species is therefore essential for maximizing methane C-H activation efficiency while suppressing over-oxidation, offering key mechanistic guidance for the rational design of selective methane oxidation catalysts.

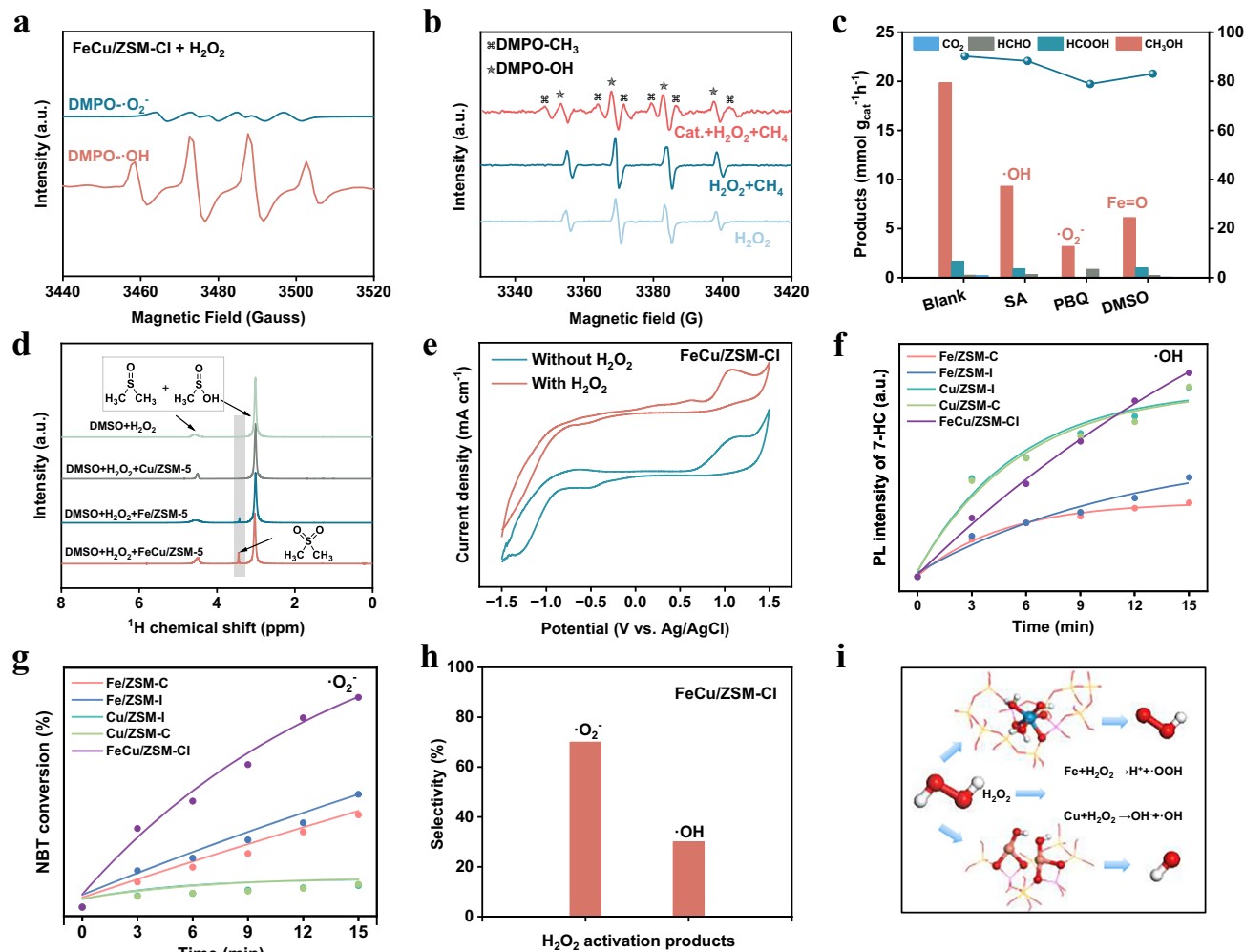

**Fig. 3 | Activation of methane and the evolution of H₂O₂ into ROS. a** EPR spectra of FeCu/ZSM-CI. **b** EPR spectra of FeCu/ZSM-CI under different reaction conditions. **c** Quenching experiments. **d** ¹H-NMR spectrum of DMSO oxidation. **e** In situ CV tests of catalysts for H₂O₂ activation. **f** PL spectra of 7-hydroxycoumarin by •OH under different catalysts. **g** NBT degradation experiment spectra by •O₂⁻ under different catalysts. **h** The ratio of •O₂⁻ and •OH generated on FeCu/ZSM-CI. **i** Decomposition pathways of H₂O₂ at FeCu/ZSM-CI metal sites.

## The influence of Fe, Cu sites on selective H₂O₂ evolution into ROS

Next, how Fe and Cu sites affect the selective generation of free radicals was carefully studied. Coumarin reacts with •OH to produce 7-hydroxycoumarin, a highly fluorescent species[44]. Plotting fluorescence luminescence (FL)-t curves can assist in semi-quantifying the Cu and Fe sites' catalyzed •OH generation rate (Fig. 3f). The catalysts containing Cu (FeCu/ZSM-CI, Cu/ZSM-I, and Cu/ZSM-C) exhibit similar PL intensity, significantly greater than that of only containing Fe catalyst (Fe/ZSM-C and Fe/ZSM-I), indicating that •OH primarily originates from H₂O₂ conversion at the Cu site. Subsequently, the production rate of •OOH was quantified by plotting the conversion of nitrotetrazolium chloride blue (NBT)-t curves (Fig. 3g)[23]. Fe containing catalyst (e.g. FeCu/ZSM-CI, Fe/ZSM-C, and Fe/ZSM-I) show the higher slope than that of only Cu containing catalyst (Cu/ZSM-I and Cu/ZSM-C), suggesting that •OOH primarily originates from the H₂O₂ conversion at the Fe site (Fig. 3h). When the ratio of Fe and Cu is adjusted from 3:1 to 1:2.5, the highest content of •OH and •O²⁻/•OOH occurs at a ratio of 1:1 (Supplementary Fig. 32).

Furthermore, the position of metal active site (e.g., inside and outside) was also important for efficient utilization of H₂O₂ and selective conversion of CH₄. When quantifying different free radicals using benzoic acid and NBT, •OH is preferentially detected

(Supplementary Fig. 32d). This is because the concentration gradient of H₂O₂ decreases from the outside to the inside during diffusion in the molecular sieve. Cu sites have a slower H₂O₂ conversion rate, but can balance •OH production in the outside with higher H₂O₂ concentration. Fe sites have a fast H₂O₂ conversion rate and efficient utilization of H₂O₂ into •O₂⁻/•OOH when in the inside with lower H₂O₂ concentration. This distribution of Fe and Cu prevents overoxidation by maintaining appropriate concentrations of •O₂⁻/•OOH and •OH.

The FeCu catalyst demonstrated exceptional H₂O₂ utilization efficiency (74.6%) (Supplementary Fig. 33), the highest among all tested catalysts. This dual-metal promoted gradient decomposition of H₂O₂ not only enhances methanol selectivity but also significantly improves the overall oxidant utilization efficiency. The synergistic effect between Fe and Cu sites facilitates a controlled decomposition pathway[45,46], minimizing unproductive H₂O₂ dissociation while maximizing its conversion to active oxygen species for selective methanol formation (Fig. 3i).

## Theoretical investigation on the evolution path from methane to methanol

In situ Diffuse Reflectance Infrared Fourier Transform Spectroscopy (in-situ DRIFTS) was employed to identify methane oxidation

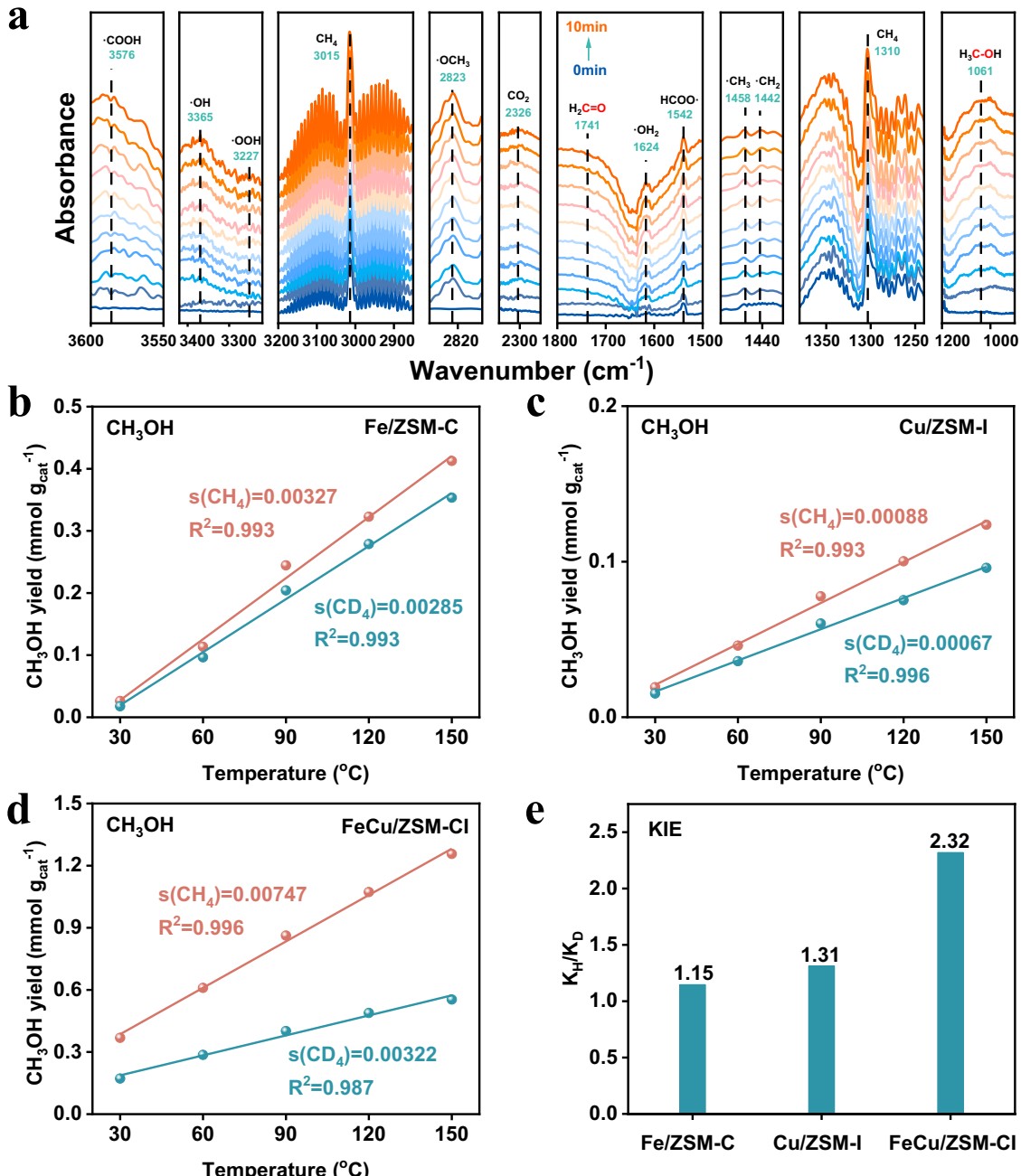

**Fig. 4 | The conversion of CH₄ to CH₃OH over FeCu/ZSM-Cl. a** In situ Diffuse Reflectance Infrared Fourier Transform Spectroscopy (in-situ DRIFTS) spectra of CH₄ on FeCu/ZSM-Cl catalyst at 80 °C. **b–d** Kinetic isotope effect experiment of CH₄ oxidation to CH₃OH over Fe/ZSM-C, Cu/ZSM-I, and FeCu/ZSM-Cl. **e** Kinetic isotope effect experiment of CH₃OH production over as-prepared catalysts.

intermediates catalyzed by FeCu/ZSM-Cl at 80 °C during selective methane conversion (Fig. 4a). After introducing water vapor with $H_2O_2$, peaks at 3365 cm⁻¹, 3227 cm⁻¹, and 1624 cm⁻¹, corresponding to •OOH, •OH, and •$OH_2$, gradually strengthened[47]. Upon adding methane continuously, the peak at 1458 cm⁻¹ belonging to •$CH_3$ became apparent along with strong characteristic peaks attributed to •$OCH_3$ and •COOH observed at 2823 cm⁻¹ and 3576 cm⁻¹ respectively[48]. As the reaction progressed, these two characteristic peaks increased less while those at 1061 cm⁻¹ corresponding to C-O bond in $CH_3OH$ continued increasing[33,48]. Associated with ¹H-NMR results that show trace amounts of $CH_3OOH$ in the product (Supplementary Fig. 34), it indicates that $CH_3OH$ is converted from the intermediates of $CH_3OOH$[22].

Identifying the rate-determining step (RDS) in a reaction is crucial for precisely enhancing the overall reaction rate. The kinetic isotope effect (KIE) of $CH_3OH$ product reveals that the $k_H/k_D$ value is 1.15 for Fe/ZSM-C and 1.31 for Cu/ZSM-I (Fig. 4b–e), both close to 1. This suggests that C-H bond activation has minimal influence on the reaction rate, confirming that $H_2O_2$ activation is the RDS for both Fe/ZSM-C and Cu/ZSM-I. In contrast, FeCu/ZSM-Cl exhibits a significantly higher $k_H/k_D$ value of 2.32, indicating that C-H bond activation plays a more dominant role in catalytic activity. This confirms that C-H bond activation is the RDS for FeCu/ZSM-Cl. A similar phenomenon was observed in the KIE (HCOOH) studies (Supplementary Fig. 35–36). Overall, due to the synergistic interaction between Fe and Cu dual single atoms, the RDS

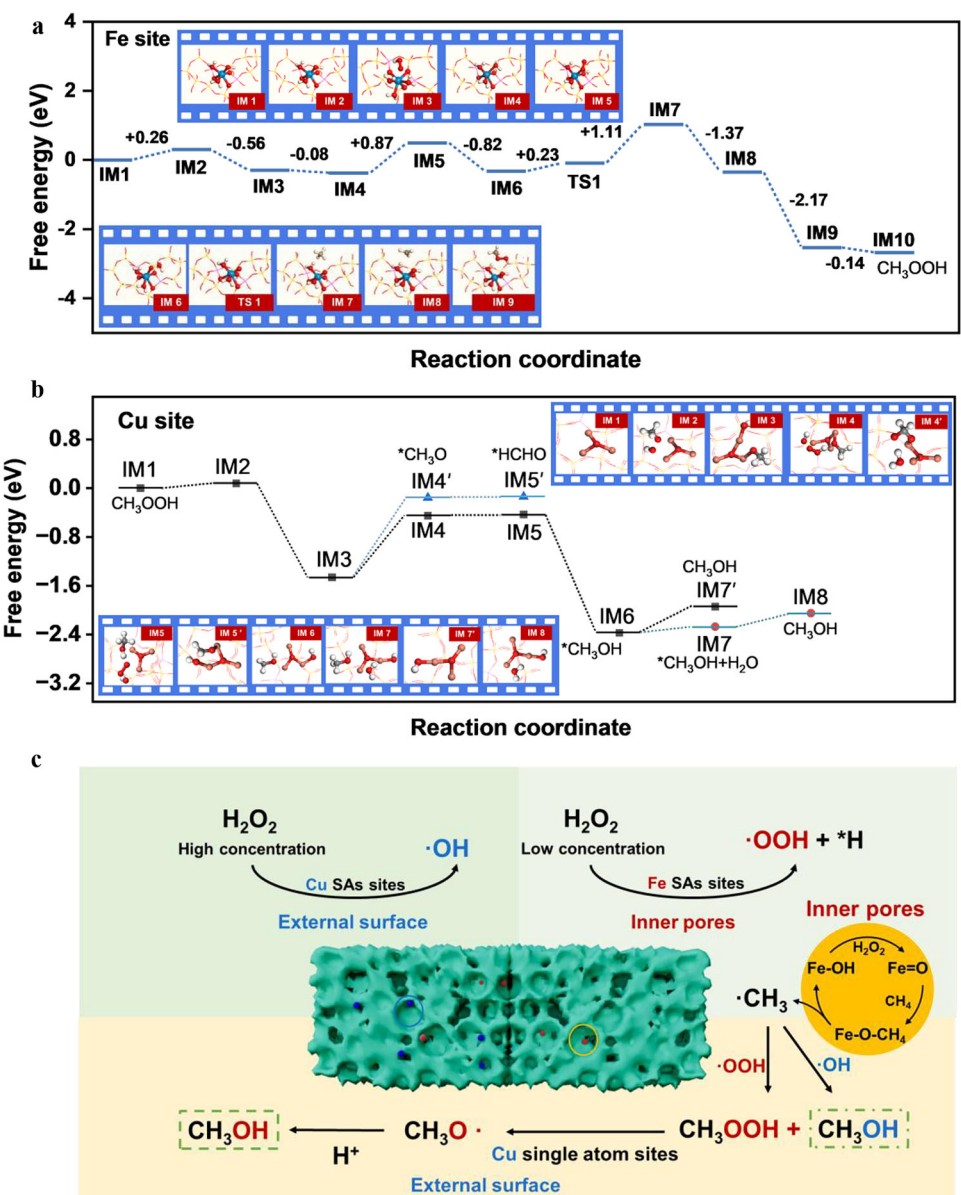

**Fig. 5 | Theoretical research on the selective production of methanol from methane. a** Energy profiles for the conversion of $CH_4$ to $CH_3OOH$ over Fe sites. **b** Energy profiles for the conversion of $CH_3OOH$ to $CH_3OH$ over Cu sites. **c** Diagram of the process of low temperature and high selectivity conversion of methane to methanol.

shifts from $H_2O_2$ activation to C-H bond activation. The enhanced generation of reactive oxygen species via $H_2O_2$ activation provides more active sites for methane oxidation, thereby boosting catalytic performance.

After confirming the important intermediates of $CH_3OOH$, DFT calculation were used to study the activation and transformation processes of $CH_4$ at functional Fe, Cu sites, respectively (Fig. 5a, b, Supplementary Figs. 37–38). The reaction can be divided into three stages due to the spatial and functional differences between Fe and Cu sites: (1) Reaction pathways and energy barrier analysis, (2) Competitive analysis of $H_2O_2$ decomposition pathways, (3) Suppression mechanism of $CH_3OH$ over-oxidation.

**Fe site pathway (Fig. 5a).** This pathway now starts with the adsorption of $H_2O_2$ at the Fe site, followed by heterolytic cleavage leading to the formation of high-valent Fe=O and •OOH species, which subsequently activate the C-H bond in $CH_4$. A comparison of the activation barriers for methane activation at Fe and Cu sites shows that the Fe site exhibits

a lower activation barrier (1.11 eV vs. 1.25 eV), indicating its superior capability for initial methane activation (Supplementary Fig. 37).

**Cu site pathway (Fig. 5b).** This pathway begins with the adsorption of the intermediate $CH_3OOH$ at the Cu site, followed by O-O bond cleavage to form •$OCH_3$, ultimately yielding $CH_3OH$. This clearly illustrates the critical role of the Cu site in the selective conversion of $CH_3OOH$ to $CH_3OH$.

To investigate the preference of $H_2O_2$ decomposition at different sites, we compared the pathways leading to •OOH and •OH formation (Supplementary Fig. 38–39). The results indicate that formation of •OOH is thermodynamically more favorable at the Fe site, while it is unfavorable at the Cu site. In contrast, the energy barrier for •OH formation at the Cu site is only 0.37 eV, making this pathway kinetically more accessible. These computational results are consistent with our EPR and radical trapping experiments (Fig. 3f–h) and confirm that Fe and Cu sites play distinct roles in $H_2O_2$ activation. Fe sites predominantly generate •OOH, whereas Cu sites favor the formation of

•OH. Meanwhile, the introduction of $H_2O$ further facilitates the activation of $H_2O_2$.

The oxidation of *$CH_3O$ by •OH to form formaldehyde has an energy barrier of 1.33 eV, which is higher than the barrier for its conversion to $CH_3OH$ (1.03 eV) (Fig. 5b). This indicates that *$CH_3O$ is less likely to undergo further oxidation to formaldehyde. Furthermore, $CH_3OH$ exhibits weak adsorption at the Cu site. The presence of water molecules further lowers the desorption energy barrier of methanol from 0.43 eV to 0.22 eV, significantly promoting its desorption from the active site into the bulk solution. This effect effectively prevents the subsequent over-oxidation of methanol at the Cu site, thereby ensuring high methanol selectivity at the mechanistic level.

In summary, by refining the depiction of the reaction pathways and incorporating energy analyses of competing reactions, which confirmed that the spatial and functional synergy between Fe and Cu sites is key to achieving efficient methane activation and high methanol selectivity (Fig. 5c).

## Discussion

This work demonstrates that spatially engineering dual single-atom Fe and Cu sites within the hierarchical pore network of ZSM-5 enables gradient-controlled $H_2O_2$ activation, allowing for both efficient C-H bond activation and selective methanol production from methane under mild conditions. By decoupling ROS evolution across pore domains high-valent Fe=O species and •OOH in the micropores for methane activation, and Cu-mediated •OH reactions on the external surface for $CH_3OH$ formation overoxidation is suppressed while maximizing $H_2O_2$ utilization. The FeCu/ZSM-CI catalyst achieves a rare combination of high activity, selectivity, and oxidant efficiency. Mechanistic insights from isotopic, spectroscopic, and theoretical studies reveal Fe-Cu synergy shifts the rate-determining step from oxidant activation to C-H activation. These findings establish spatial site isolation as a general and scalable strategy for regulating ROS pathways in alkane oxidation, offering new design principles for sustainable catalytic conversions of inert molecules.

## Methods
### Materials and chemicals
Iron acetylacetonate ($Fe(C_5H_7O_2)_3$, AR, 99%, Macklin) and Copper (II) nitrate Gerhardite ($Cu(NO_3)_2$ 3$H_2O$, AR, 99%, Macklin) were used as metal precursors. Sodium aluminum oxide ($Al_2Na_2O_4$, AR, 98%, Macklin), Sodium hydroxide (NaOH, 95%, graininess, Macklin) Silicasol (Silica content: 29-31%, Macklin) and Ammonium chloride ($NH_4Cl$, AR, 99.99%, Macklin) were used as a raw material for the synthesis of H-ZSM-5. Hydrogen peroxide ($H_2O_2$, AR, 30%, Macklin) was used as the oxidizing agent. Methane (99.999 vol.%, Qingdao Xin ke yuan) was used as the feedstock gases. All chemicals were used as received without any further purification. Deionized water was used throughout the research.

### Catalyst synthesis
Fe acetylacetone was chosen as the source to incorporate Fe into inner small pores due to its similar size and ability to be confined in the micropores. The stable coordination structure of Fe acetylacetone reduces its reaction with Si and Al sources, while the residual Fe acetylacetone in the external large pores can be eliminated by washing away because of its slightly solubility in water. Fe species were grown in-situ using the crystal seed method and then calcined to produce Fe/ZSM-C. Cu species were added through impregnation, resulting in FeCu/ZSM-CI, where C and I denote the sequence of crystal seed method and impregnation method for Fe and Cu respectively. To conduct control experiments, a series of catalysts decorated with different combinations of metals (e.g., Fe/ZSM-C, Cu/ZSM-I, CuFe/ZSM-CI) were obtained by varying metal types, ratios, and introduction methods.

Initially, Fe site was grown in situ using the crystal seed method to fix it in the inner small pores of ZSM-5. To achieve this, Fe acetylacetone was chosen as its size is similar to that of the micropore. Residual Fe present in external large pores (such as mesoporous) was mostly eliminated by washing with water after hydrothermal synthesis. The above precursor was then obtained through calcination and named Fe/ZSM-C (C refers to crystal seed method). Next, $Cu(NO_3)_2$ solution was used for impregnation on Fe/ZSM-C under controlled conditions resulting in a catalyst called FeCu/ZSM-CI (I stands for impregnated method).

**Synthesis of Fe/ZSM-C.** Fe/ZSM-C was synthesized using crystal seed method. This synthetic procedure comprised two steps: the synthesis of ZSM-5 seeds and the synthesis of Fe-ZSM-5.

In the first step, 15 g of Silica sol was dissolved in 7 mL of a NaOH solution (1 mol $L^{-1}$) under constant stirring at 100 °C for 1 h. Simultaneously, a solution was prepared by dissolving 0.45 g of sodium aluminate oxide in 7 mL of a NaOH solution (1 mol $L^{-1}$). The amalgamation of these two solutions resulted in the formation of a synthetic aluminosilicate gel with a molar composition of 4 $Na_2O$: 1 $Al_2O_3$: 36 $SiO_2$: 460 $H_2O$. The gel underwent stirred vigorously at 100 °C for 2 h and was then transferred to a stainless-steel autoclave for crystallization at 180 °C for 48 h. The resultant crystals were collected by filtration, cleaned with deionized water until the pH of the filtrate is neutral.

Subsequently, the obtained crystals were dried at 100 °C to produce the ZSM-5 seeds. In the second step, a synthetic aluminosilicate gel was prepared using the same method as described in the first step. 2 mL of an aqueous solution containing 0.21 g of iron acetylacetonate and 0.06 g of seeds were added sequentially to the gel, followed by 30 min of stirring. The mixture was then transferred to an autoclave for crystallization at 180 °C for 48 h. After crystallization, the hydrothermal catalyst was placed in a beaker, to which 200 mL of deionized water was added. The mixture was at 80 °C for 12 h to remove the residual iron acetylacetone. The catalyst was washed and filtered with a large amount of deionized water until the filtrate presents a clear color, and the present of Fe ions are determined using KSCN. Washing and filtration were stopped when no iron acetylacetone was detected in the filtrate. The filter cake was dried at 80 °C to obtain the synthesized Fe/ZSM-C.

Finally, Fe/ZSM-C was converted to H-type Fe/ZSM-C through ion exchange with ammonium chloride at 80 °C, followed by calcination in air at 550 °C for 6 h. The Fe loading was determined by ICP-OES to be approximately 0.6 wt%. In the synthesis of H-ZSM-5, the only deviation from the standard procedure was the omission of iron acetylacetone in the second step, while all other steps and conditions remained unchanged. The obtained H-ZSM-5 was used in the following other catalyst synthesis.

**Synthesis of Cu/ZSM-I.** H-ZSM-5, synthesized using the seed method above, was used as the molecular sieve carrier. To prepared the catalyst, 1 g of synthetic H-ZSM-5 was suspended in 50 mL deionized water and fully stirred to obtain solution A. A certain amount of Copper (II) nitrate was dissolved in 10 mL deionized water, followed by ultrasonic treatment for 5 min to ensure complete dissolution. The pH value of the solution was then adjusted to 3 - 4. The prepared copper nitrate solution was slowly added into solution A at a rate of 0.5 mL $min^{-1}$, followed by continuous stirring and impregnation for 24 h. After impregnation, the solution was washed and filtered with a large amount of deionized water. The obtained catalyst was dried at 80 °C, and then calcined at 550 °C in the Muffle furnace for 6 h to obtain Cu/ZSM-I.

**Synthesis of FeCu/ZSM-CI.** FeCu/ZSM-CI was prepared by introducing copper into Fe/ZSM-C as a precursor by the impregnation

adsorption method. The procedure can be divided into two main steps. Take the metal load Fe / Cu (0.6 wt% / 0.6 wt%) as an example: First, the precursor Fe/ZSM-C was synthesized following the same method as described for Fe/ZSM-C above. After that, 1 g of the synthesized Fe/ZSM-C was placed in 50 mL deionized water and stirred thoroughly to obtain solution B. Then 0.02 g of Copper (II) nitrate was dissolved in 10 mL deionized water and ultrasonically treated for 5 min to ensure complete dissolution. The pH of the solution to 3–4. The prepared copper nitrate solution was added into B solution at the rate of 0.5 mL/min, followed by stirring and impregnation for 24 h. After impregnation, the solution was washed and filtered with a large amount of deionized water. The obtained catalyst was dried at 80 °C and then calcined at 550 °C in muffle furnace for 6 h to obtain FeCu/ZSM-CI.

**Synthesis of FeCu/ZSM-CC.** The synthesis procedure is the same as Fe/ZSM-C, but the difference is that Iron acetylacetonate and Copper (II) nitrate are added at the same time.

**Synthesis of FeCu/ZSM-IC.** The precursor of Cu-ZSM-5-C was synthesized using Copper (II) nitrate instead of Iron acetylacetonate, followed by the adsorption and impregnation of Fe. The procedure follows the same steps as described above.

**Synthesis of FeCu/ZSM-II.** H-ZSM-5, synthesized using the seed method described above, was used as the molecular sieve carrier. 1.00 g of synthetic H-ZSM-5 was placed in 50 mL deionized water and stirred thoroughly to obtain solution C. A specific amount of Iron acetylacetonate and Copper (II) nitrate were dissolved in 10 mL deionized water, then ultrasonicated for 5 min to ensure complete dissolution. The pH of the solution was adjusted to 3 - 4. The prepared copper nitrate solution was then added uniformly into solution A at 0.5 mL min$^{-1}$, followed by stirring and impregnation for 24 h. After impregnation, the solution was washed and filtered with a large amount of deionized water. The resulting catalyst was dried at 80 °C and then calcined at 550 °C in the muffle furnace for 6 h to obtain FeCu/ZSM-II.

**Catalyst testing.** The selective oxidation of methane experiment was carried out in a 50 mL high-pressure reactor. The catalyst (5 mg) was uniformly dispersed in 20 mL of distilled water and sonicated for 15 min. A specific amount of $H_2O_2$ was added and the reactor was sealed. The reactor was purged with argon gas 3 - 5 times to replace the air. Then, methane was injected into the reactor to reach the required pressure. The reaction was carried out in an oil bath for 3 h. After the reaction, the reactor was cooled to below 10 °C using an ice bath, and both the gas and liquid were collected.

**Catalyst characterization**
X-ray Diffraction (XRD) was performed by the diffractometer (X'Pert PRO MPD, PANalytical, Netherlands) with Cu Kα radiation (40 kV, 100 mA, λ = 0.154 nm). The morphology of materials was observed by Scanning electron microscope (SEM), whose model is JSM-7500F scanning electron microscopes (Japan). High-Resolution Transmission Electron Microscopy (HRTEM) and Energy Dispersive X-ray Spectroscopy-mapping (EDS-mapping) images were captured using Tecni G30 instrument (FEI, USA). The morphology of the samples was further observed by Aberration Corrected High-Angle Annular Dark Field Scanning Transmission Electron Microscope (AC-HAADF-STEM, Themis Z, Thermo Scientific, USA). UV-Vis diffuse reflectance spectra were obtained from the spectrometer (UV-2700, Shimadzu, Japan) furnished with an integrating sphere device. Solid-state $^{27}Al$ Magic-Angle Spinning NMR ($^{27}Al$ NMR MAS) cross polarization spectroscopy was measured on a JEOL ECA-600 spectrometer at a resonance frequency of 156.4 MHz using a 4 mm sample rotor with a spinning rate of

15.0 kHz. The $^{27}Al$ chemical shift was referenced to -0.54 ppm of $AlNH_4(SO_4)_2 \cdot 12H_2O$. The information on the electronic states of the material surface was collected via the X-ray Photoelectron Spectrometer (XPS, ESCALAB 250Xi, Thermo Scientific, USA).

The metal site structure of the catalyst materials was determined by X-ray Absorption Spectroscopy (XAS), whose date for Fe K-edge and Cu K-edge were collected at the 1W1B station of the Beijing Synchrotron Radiation Facility (BSRF), where the storage rings operated at 2.5 GeV with a maximum current of 250 mA. For Fe-containing and Cu-containing references (i.e., Fe foil, FeO, $Fe_2O_3$, Cu foil, $Cu_2O$, and CuO), data were collected in transmission mode using an ionization chamber, while for Fe-containing zeolites and Cu-containing zeolites, data were obtained in fluorescence excitation mode using a Lytle detector. The X-ray Absorption Near Edge Structure (XANES) and Fourier-transformed Extended X-ray Absorption Fine Structure (EXAFS) data were analyzed using ATHENA and ARTEMIS software, respectively, and MATLAB software was employed for the analysis of wavelet-transformed EXAFS data.

Electron paramagnetic resonance (EPR) spectrometer was used to characterize radicals in the reaction. EPR spectra were measured on CIQTEK EPR200M with continuous-wave X band frequency, 5,5-dimethyl-1-pyrroline-N-oxide (DMPO) as the radical trap. The samples were dispersed in water-dissolved $CH_4$ and $H_2O_2$ to detect •$CH_3$ and •OH, and also in methanol-dissolved $H_2O_2$ to detect •$O_2^-$.

In-situ Diffuse Reflectance Infrared Fourier Transform Spectroscopy (DRIFTS) measurements were measured on the instrument (VERTEX70, Bruker, Germany), the mercury cadmium telluride (MCT) detector was adopted, and Ar was bubbled into $H_2O_2$ when the temperature was raised and stabilized to 80 °C. The background correction was stabilized, and $CH_4$ replaced Ar for testing.

Inductively Coupled Plasma Atomic Emission Spectroscopy (ICP-AES, Agilent 730, USA) was used to determine the metal contents. Temperature-Programmed Desorption measurements were carried out on Micromeritics AutoChemHP-2950. The Fluorescence Spectra were collected by the fluorescence spectrophotometer (RF-6000, Shimadzu, Japan).

The gas chromatography (Scion 456 C, Tianmei, China) is equipped with a thermal conductivity detector (TCD), two flame ionization detectors (FID), a methanizer, and a headspace autosampler (DK-5001A, Beijing Zhongxing, China), were used to quantify gaseous and $CH_3OH$ and $CO_2$ products. High-performance liquid chromatography (HPLC, Prominence-i, LC-2030 Plus, Japan) equipped with a Refractive Index Detector (RID) was used to quantify HCOOH products. UV-Vis diffuse reflectance spectra (UV-2700, Shimadzu, Japan) was used to quantify HCHO products.

This combination of sophisticated techniques provided comprehensive information about the structure, composition, and catalytic behavior of the materials under study.

**Low temperature selective oxidation of methane.** The methane carbonylation experiment was carried out in a 50 mL high-pressure reactor (Shi ji shen lang). The catalyst (5 mg) was uniformly dispersed in 20 mL of distilled water and sonicated for 15 min. A certain amount of $H_2O_2$ was added and the reactor was sealed. The reactor was purged with argon gas to replace the air for 3–5 times. Then, methane was injected at the required pressure. The reaction was carried out in an oil bath for 3 h. After the reaction, the reactor was cooled to below 10 °C in an ice bath, and the gas and liquid were collected.

**Cyclic experiment.** The recycle test followed the same procedure. After each run, the spent catalyst was separated, washed with a large amount of $H_2O$ then dried at 80 °C in the vacuum oven for the next cycle. The same amount of catalyst was used and repeated experiments were carried out under the same experimental conditions to verify the stability of the catalyst.

## Data availability

The raw data generated in this study are provided in the Supplementary Information. All data are available from the corresponding author upon request. Source data are provided with this paper.

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

## Acknowledgements

This work was supported by National Natural Science Foundation of China (22322815, 22561142234, 22179146, and 22138013), Shandong Provincial Natural Science Foundation (Grant No. ZR2025QA14), the Fundamental Research Funds for Central Universities (18CX07009A), Independent Innovation Research Project (Science and Engineering) (20CX06072A). The National Key Research and Development Program of China (2023YFB4104500, 2023YFB4104503), the Key Research and Development Program of Shandong Province (2024ZLGX08), and the Science and Technology Innovation Project of the Shandong Energy Group Co., Ltd. (SNKJ2023A03). We thank Dr. Shaodong Sun of Research Center on Advanced Chemical Engineering and Energy Materials, China University of Petroleum (East China) for the technical support and valuable help of liquid chromatography-mass spectrometry (LC-MS).

## Author contributions

M.W. and W.W. conceived and supervised the project. H.Z., S.W., and W.W. conducted most experiments including synthesis, characterization and testing, as well as data analysis. Y.L. (Yang Li) carried out DFT calculations section. H.Q. performed characterization and testing analysis. M.W. conducted spectrum analysis. Q.C. and B.Z. contributed to data analysis of the EPR spectra. S.Z. conducted DRIFTS analysis. P.Z. and C.G. contributed to data analysis of the X-ray absorption spectroscopy. Y.L. (Yunyun Li) conducted an analysis of the mass spectrometry. Q.H. and M. W. provided advice and expertise. H.Z., S.W., and W.W. wrote and revised the paper. All authors discussed the paper.

## Competing interests

The authors declare no competing interests.
