## [Transparent Peer Review file · Nature Communications]

FeCu Dual-Single-Atom Catalyst Promotes Gradient H₂O₂ Activation for Enhanced Methane Oxidation to Methanol

Corresponding Author: Professor Wenting Wu

Version 0:

Reviewer comments:

Reviewer #1

(Remarks to the Author)

The paper reports the oxidation of methane in aqueous conditions using a CuFe/zeolite catalyst and H₂O₂. The authors report high methanol selectivity ~90% and productivity of ~ 20 mmol / g/ h, which is largely in line with many previous studies of these systems e.g. DOI: 10.1002/ange.201108706.

The authors state that these results are significant due to the nature of the catalyst preparation with spatially separated Fe and Cu species. There is quite some confusion throughout the paper on the description of these sites. Throughout the paper, this is described as i) Fe and Cu spatially separated ii) due to Fe-Cu interactions iii) FeCu metal species iv) Fe-Cu synergy etc

The mechanistic conclusions also largely mirror those of the papers cited above from 2012 “Hydrogen peroxide reacts at the iron centres to produce species capable of the activation of the carbon–hydrogen bond, forming methyl hydroperoxide as the primary product. Whilst copper does not play a direct role in methane activation, it facilitates the formation of methanol by inhibiting over oxidation to formic acid and CO₂.”

This means that I interpret this paper as an incremental step in catalyst preparation by optimization of loading, location and speciation of metal sites already proven to be an active system for this reaction.

General Comments

The paper seems to have some errors – e.g. figure 2 caption does not match figure, and some text appears to be missing lines 213 – 215.

The mechanistic studies are sometimes confusing – EPR shows the presence of OH, superoxide and methyl radicals, then it is stated that the production rate of OOH by Cu was measured by conversion of (NMT)-t curves. OOH is not cited in the proposed mechanism. Did the authors consider the half-life of the DMPO-OOH adduct? Which then degrades into DMPO-OH? Further, it is then stated that the methyl radicals “accumulate and react with a significant number of OOH species along with a smaller quantity of OH) which is not represented in the proposed mechanism

Reviewer #2

(Remarks to the Author)

This work represents an advancement in catalytic methane oxidation, achieving a remarkable combination of high methanol yield (20.20 mmol g⁻¹ h⁻¹), exceptional selectivity (90.1%), and remarkable H₂O₂ utilization efficiency (74.6%) through innovative spatial engineering of dual Fe-Cu single-atom sites in ZSM-5. The decoupling of reactive oxygen species generation across pore domains, high-valent Fe=O in micropores for C-H activation, and surface Cu for selective methanol formation, opens avenues for sustainable hydrocarbon valorization with atom-economical oxidant use. I recommend accepting the manuscript after minor revisions and addressing the following points:

1. The authors have confirmed the heterogeneous nature of the catalytic system through hot filtration experiments and ICP analysis of the catalyst before and after reaction. To further validate these findings, we recommend conducting the following control experiment: exposing the hot-filtered filtrate to fresh ZSM-5 and monitoring any changes in reaction activity. This

approach would help distinguish between genuine heterogeneous catalysis and potential contributions from soluble active species.

2. ¹³C NMR characterization of the reaction products is necessary to unambiguously confirm methanol formation.

3. The authors achieved remarkably high H₂O₂ utilization efficiency and methanol selectivity through precise control of metal spatial arrangement and extensive comparative experiments - a highly interesting finding. To further elucidate the structural effects, it would be valuable to examine whether physical mixing of Fe/ZSM-C and Cu/ZSM-I catalysts could produce similar promotional effects. Such verification would provide more compelling evidence for the proposed structure-function relationship."

4. Regarding the reaction mechanism, the authors have convincingly demonstrated through CV measurements the in situ formation of high-valent metal species during catalysis - an important finding. To further strengthen the experimental validation, it would be essential to include control CV tests of the bare ZSM-5 support. This comparative analysis would unambiguously confirm that the observed redox peaks originate from the metal active centers rather than the support material itself.

5. The authors are suggested to improve the methodological description of the radical trapping experiments (particularly the details for •CH₃, •O₂⁻, and •OH trapping), which would significantly enhance the reproducibility and reference value of the study.

Reviewer #3

(Remarks to the Author)

This work synthesized spatially separated FeCu/ZSM-CI catalysts, with Fe located in the inner channels and Cu on the external surface, enabling efficient direct methane to methanol by using H₂O₂ as the oxidizing agent. Interestingly, it was proposed that the tandem conversion mechanism towards methanol production was responsible for high methanol selectivity. The mechanism was further confirmed by the comparison with Fe-, Cu-, and FeCu-contact-ZSM catalysts. The paper might be publishable in Nature Communications, but the authors must address the following issues first:

1. The water is of vital importance for methane activation and methanol production using hydrogen peroxide as the oxidizing agent. Not only could it accelerate the methane activation and H₂O₂ decomposition as a co-catalyst, but also the hydrophobic effect of the channel could significantly affect the distribution of methane and hydrogen peroxide, affecting methanol selectivity. The implicit model was used in the calculations for the solvation effect for this manuscript, which could not reflect the abovementioned key roles of water. The details should be included.

2. The H₂O₂ self-decomposition is one key problem when using H₂O₂ as an oxidizing agent. The lower H-O bonding of H₂O₂ results in the preference of H₂O₂ activation and self-decomposition generally results in the lowered methane conversion (Chin. J. Catal. 51(2023), 135; Ind. Eng. Chem. Res. 63(2024), 11384). The authors need to offer more data about H₂O₂ self-decomposition.

3. Direct methane to methanol (DMTM) generally faces the conversion-selectivity trade-off (ACS Catal. 8(2018), 6894). The high selectivity could be easily obtained by the lower conversion that is generally lower than 20%, even though the methane conversion is not prior to methanol deep oxidation (Angew. Chem. Int. Ed. 56(2017), 16464). The conversion data should be provided.

4. How were the models for DFT simulations established? Why were T8 and T12 sites chosen as the acid sites? Why were the iron mononuclear sites coordinated with six oxygen and Cu binuclear sites chosen as the models? In particular, small Cu particles or clusters would be located on the external surface when following the experimental observation in the manuscript.

5. The reaction mechanism should be further checked. It is unreasonable for the energy profiles in Fig. 5, starting with the transition states of TS0 and TS1. Moreover, the side reaction including H₂O₂ decomposition and deep oxidation of CH₃OH and CH₃OOH should be included to confirm the proposed mechanism towards high selectivity of methanol.

6. It was suggested that iron species are in the direct channel for Fe/ZSM-C and FeCu/ZSM-CI. However, it is difficult to find the bright spots for iron species in the direct channel in Figs. 2a and 2c.

7. In addition, FeCu/ZSM-CI catalysts with Fe located in the inner channels and Cu on the external surface was characterized after the catalyst synthesis. However, the metallic center is easily to be detached from the zeolite framework during the reaction, leading to the lowered performance. The longer period, the stability test, and corresponding catalyst characterization should be provided to confirm the active sites and reaction mechanism.

8. Due to the high cost of H₂O₂, it is unable to be applied in the realistic catalyst system using H₂O₂ as the oxidizing agent for DMTM. Is it possible to extend the current system to the DMTM using O₂?

Version 1:

Reviewer comments:

Reviewer #1

(Remarks to the Author)

I thank the authors for making substantial changes and clarifications to the manuscript based on the comments of the three reviewers.

While the authors do emphasize the higher performance of this material compared to earlier studies it is iterative in that its is an optimization of a known material - the new configuration of the active sites reveals some similar and some different mechanistic aspects to known studies - but is not a fundamental step change in terms of discovery - apart from in terms of

activity through active site optimization - particularly methane conversion.

Can the authors further clarify their new mechanism where they propose a Fe-O-CH₄ in the mechanism figures? What experimental evidence is there for this?

Reviewer #2

(Remarks to the Author)

The authors have addressed all the concerns proposed by the reviewer. Hence, I recommend publishing this manuscript as it is without further revision.

Version 2:

Reviewer comments:

Reviewer #1

(Remarks to the Author)

Comments have been addressed

REVIEWER COMMENTS

Reviewer #1:

The paper reports the oxidation of methane in aqueous conditions using a CuFe/zeolite catalyst and H₂O₂. The authors report high methanol selectivity ~90% and productivity of ~ 20 mmol / g/ h, which is largely in line with many previous studies of these systems e.g. DOI: 10.1002/ange.201108706.

We thank the reviewer for raising this point. While our work indeed builds upon the foundational discovery by Hutchings et al. regarding the Fe-Cu synergy in ZSM-5 for methane oxidation, our study represents a significant step forward, as demonstrated by the following key advances:

We deliberately constructed a spatially segregated dual-metal architecture (Fe inside, Cu outside), which differs distinctly from the previously proposed binuclear Fe species and physically separable Cu sites. We uncovered a novel mechanism of gradient H₂O₂ activation and high-valent Fe=O catalysis, in which copper actively converts the intermediate CH₃OOH into CH₃OH, thereby enhancing methanol selectivity, and achieved a remarkable improvement in methanol yield (20.20 mmol g⁻¹ h⁻¹) and H₂O₂ utilization efficiency (74.6%), successfully addressing critical challenges in this field.

Through experimental investigation, we observed a shift in the rate-determining step due to enhanced metal synergy: over Fe/ZSM-5, the rate-determining step is H₂O₂ activation, whereas over FeCu/ZSM-5, it shifts to methane C-H bond activation. Therefore, we believe our work provides a new design principle for manipulating reactive oxygen species via spatial control of active sites, offering groundbreaking fundamental insights and a superior catalytic system that exceeds the current state of the art.

1. The authors state that these results are significant due to the nature of the catalyst preparation with spatially separated Fe and Cu species. There is quite some confusion throughout the paper on the description of these sites. Throughout the paper, this is described as i) Fe and Cu spatially separated ii) due to Fe-Cu interactions iii) FeCu metal species iv) Fe-Cu synergy etc

We sincerely thank you for this exceptionally precise and important comment. You

are absolutely correct that the description of the active sites throughout our manuscript was indeed inconsistent and conceptually confusing, which undoubtedly affected the paper's clarity and logical rigor. We apologize for the confusion this has caused.

Following your suggestion, we have systematically revised the entire manuscript to unify and clarify the relevant concepts. We have clarified that "Fe and Cu spatially separated" refers to the macroscopic distribution of the physical locations of the Fe and Cu atoms, where Fe species are anchored within the internal channels of ZSM-5 via the crystal seed method while Cu species are located on the external surface via impregnation, representing the fundamental design and static structural feature of the catalyst. Meanwhile, "synergy" describes how Fe and Cu, despite their spatial separation, work together functionally through a continuous, tandem reaction pathway to achieve high performance.

To ensure conceptual consistency, we have revised or removed ambiguous expressions throughout the text, such as "FeCu metal species" and "FeCu interactions," and now consistently use "FeCu synergy" to describe their cooperative effect. When describing the static catalyst structure, we uniformly employ the term "spatially separated."

2. The mechanistic conclusions also largely mirror those of the papers cited above from 2012 "Hydrogen peroxide reacts at the iron centres to produce species capable of the activation of the carbon-hydrogen bond, forming methyl hydroperoxide as the primary product. Whilst copper does not play a direct role in methane activation, it facilitates the formation of methanol by inhibiting over oxidation to formic acid and CO₂." This means that I interpret this paper as an incremental step in catalyst preparation by optimization of loading, location and speciation of metal sites already proven to be an active system for this reaction.

We sincerely thank the reviewer for this insightful comment. We acknowledge the similarity in mechanistic description between our conclusions and the pioneering 2012 study, particularly regarding iron centers activating C-H bonds to generate methanol as the primary product, and copper's role in suppressing over-oxidation while promoting methanol formation. However, our work provides crucial mechanistic advances and

new physical insights that significantly deepen this understanding. In practice, different iron and copper species need to be selected to promote the gradient conversion of H_2O_2 depending on its concentration, a aspect not thoroughly investigated in the 2012 study. Extraframework single-atom dispersed iron tends to form $\text{Fe}=\text{O}$ species, whereas iron oxide clusters or other structures readily decompose H_2O_2 into hydroxyl radicals or oxygen, failing to effectively utilize low-concentration H_2O_2 within the pores. In contrast, for high-concentration H_2O_2 outside the pores, mild copper-based catalysts, specifically in the form of copper oxide clusters, can be employed; otherwise, H_2O_2 decomposition into oxygen is also likely to occur.

1. Spatial localization of active sites

While previous studies proposed Fe-Cu cooperation, we provide direct experimental evidence through advanced characterization techniques for their spatial separation within the zeolite framework. We demonstrate that Fe species predominantly occupy internal channels while Cu species enrich the external surface - a spatial configuration that enables gradient H_2O_2 activation and high utilization efficiency.

2. Identification of the rate-determining step shift

Our kinetic isotope effect studies reveal a fundamental shift in the catalytic cycle: while H_2O_2 activation is rate-determining for monometallic Fe catalysts, C-H bond activation becomes rate-determining in the spatially separated FeCu/ZSM-CI system. This demonstrates that Fe-Cu synergy not only affects selectivity but fundamentally alters reaction kinetics by enhancing H_2O_2 activation efficiency.

3. Analysis of reactive oxygen species

Through quantitative analysis of reactive oxygen species, we demonstrate the distinct roles of Fe and Cu sites in ROS generation Fe sites preferentially produce $\bullet\text{OOH}$ species while Cu sites generate $\bullet\text{OH}$ radicals. This spatial and functional differentiation creates a gradient ROS environment that maximizes methane activation while minimizing over-oxidation.

4. Tracking of reaction intermediates

By combining in-situ DRIFTS and NMR, we directly track the formation of CH_3OOH at internal Fe sites and its diffusion to external Cu sites, providing

experimental validation of the tandem reaction mechanism.

5. Theoretical validation of the mechanism

Our DFT calculations explicitly simulate the spatial separation effect, demonstrating how the H_2O_2 concentration gradient and different activation barriers at Fe and Cu sites collectively create an optimized reaction environment unattainable with randomly distributed metals.

In summary, while building upon the fundamental concept of Fe-Cu synergy, our work provides substantial mechanistic insights through multiple experimental approaches that significantly advance the understanding of this catalytic system.

3. The paper seems to have some errors - e.g. Figure 2 caption does not match figure, and some text appears to be missing lines 213 - 215.

Thank you for the valuable feedback. We apologize for the oversight and have revised the labeling in Figure 2 accordingly.

Fig. 2 | Catalyst model. (A-C) iDPC-STEM images of Fe/ZSM-C, Cu/ZSM-I, and FeCu/ZSM-CI. (D-F) Models of Fe/ZSM-C, Cu/ZSM-I, and FeCu/ZSM-CI catalysts.

4. The mechanistic studies are sometimes confusing - EPR shows the presence of OH, superoxide and methyl radicals, then it is stated that the production rate of OOH by Cu was measured by conversion of (NMT)-t curves. OOH is not cited in the proposed mechanism. Did the authors consider the half-life of the DMPO-OOH adduct? Which then degrades into DMPO-OH? Further, it is then stated that the methyl radicals

“accumulate and react with a significant number of OOH species along with a smaller quantity of OH) which is not represented in the proposed mechanism.

We thank the reviewer for their valuable comments regarding the clarity of the reaction mechanism description, particularly concerning the identification of different radical species ($\bullet\text{OH}$, $\bullet\text{OOH}$, $\bullet\text{O}_2^-$), their roles in the reaction, and potential interference in the corresponding EPR tests. We fully agree that the original descriptions could indeed cause confusion. Below, we provide point-by-point clarifications addressing the specific issues raised.

1. Detection, Half-Life, and EPR Characterization of $\bullet\text{OOH}$

You have correctly highlighted the complexity associated with the $\bullet\text{OOH}$ radical and its DMPO adduct (DMPO-OOH) in EPR testing. The DMPO-OOH adduct is indeed unstable in aqueous solution and tends to degrade into the DMPO-OH adduct, which may lead to the absence or masking of the $\bullet\text{OOH}$ signal in EPR spectra.

Fig. 3 | (A) EPR spectra of FeCu/ZSM-Cl.

In acidic or neutral aqueous solutions, $\bullet\text{O}_2^-$ rapidly protonates to form $\bullet\text{OOH}$. In our experiments, the detection of DMPO- $\bullet\text{O}_2^-$ adducts was conducted in methanol rather than aqueous solution, ensuring the reliability of the data, while the DMPO-OH measurements were performed in aqueous media (Fig. 3A). Given the instability of DMPO-OOH noted by the reviewer, we employed multiple complementary methods to cross-validate the presence and origin of $\bullet\text{O}_2^-/\bullet\text{OOH}$:

Indirect Quantification of $\bullet\text{O}_2^-/\bullet\text{OOH}$:

The nitroblue tetrazolium (NBT) conversion assay was used to semi-quantitatively evaluate the generation rate of $\bullet\text{OOH}$, as this reaction is unaffected by DMPO-OOH

degradation. As shown in Fig. 3G, Fe-containing catalysts (Fe/ZSM-C, Fe/ZSM-I, and FeCu/ZSM-CI) exhibited significantly higher NBT conversion rates than the Cu-only catalyst (Cu/ZSM-I), providing strong evidence that $\bullet\text{O}_2^-/\bullet\text{OOH}$ primarily originates from H_2O_2 activation at Fe sites.

Fig. 3 | (G) NBT degradation experiment spectra by $\bullet\text{O}_2^-$ under different catalysts.

Indirect Quantification of $\bullet\text{OH}$:

Similarly, the generation of $\bullet\text{OH}$ was independently assessed using a coumarin probe, which reacts with $\bullet\text{OH}$ to form the highly fluorescent product 7-hydroxycoumarin (Fig. 3F). The results confirmed that Cu sites are the main source of $\bullet\text{OH}$.

Fig. 3 | (F) PL spectra of 7-hydroxycoumarin by $\bullet\text{OH}$ under different catalysts.

In summary, although EPR spectra may show signal ambiguities due to the degradation of $\text{DMPO}\text{-}\bullet\text{O}_2^-$ adducts, the NBT degradation assay provides robust evidence that $\bullet\text{O}_2^-/\bullet\text{OOH}$ is a crucial reactive oxygen species in the reaction system, predominantly generated at Fe sites.

2. Clarification on the Reaction Mechanism of $\bullet\text{CH}_3$ with $\bullet\text{OOH}/\bullet\text{OH}$

We apologize for the lack of clarity in the original manuscript regarding the key step describing the reaction of $\bullet\text{CH}_3$ with $\bullet\text{OOH}/\bullet\text{OH}$, which was not explicitly illustrated in the mechanistic diagram (Fig. 5C). We have revised the diagram in the updated manuscript to clearly depict the reaction pathway of $\bullet\text{CH}_3$ with both $\bullet\text{OOH}$ and $\bullet\text{OH}$.

Fig. 5 | (C) Diagram of the process of low temperature and high selectivity conversion of methane to methanol.

Primary Pathway: In the internal pores of ZSM-5, $\bullet\text{CH}_3$ generated by CH_4 activation at $\text{Fe}=\text{O}$ sites preferentially react with $\bullet\text{OOH}$ to form the key intermediate CH_3OOH . This is inferred as the dominant pathway due to the high local concentration of $\bullet\text{OOH}$ (as confirmed by NBT experiments) and spatial confinement.

Minor Pathway: A fraction of $\bullet\text{CH}_3$ radicals may diffuse and react with $\bullet\text{OH}$ in the solution phase to directly form CH_3OH .

Subsequent Transformation: The CH_3OOH intermediate diffuses to the external Cu sites, where it is selectively decomposed into the final product, CH_3OH .

Summary

We once again thank the reviewer for their insightful comments, which have helped us identify and address ambiguities in the mechanistic description and illustrations. We believe the revisions have significantly improved the rigor and clarity of our mechanistic interpretation. We hope these explanations and modifications

satisfactorily address the reviewer's concerns.

Reviewer #2 (Remarks to the Author):

This work represents an advancement in catalytic methane oxidation, achieving a remarkable combination of high methanol yield (20.20 mmol g⁻¹ h⁻¹), exceptional selectivity (90.1%), and remarkable H₂O₂ utilization efficiency (74.6%) through innovative spatial engineering of dual Fe-Cu single-atom sites in ZSM-5. The decoupling of reactive oxygen species generation across pore domains, high-valent Fe=O in micropores for C-H activation, and surface Cu for selective methanol formation, opens avenues for sustainable hydrocarbon valorization with atom-economical oxidant use. I recommend accepting the manuscript after minor revisions and addressing the following points:

1. The authors have confirmed the heterogeneous nature of the catalytic system through hot filtration experiments and ICP analysis of the catalyst before and after reaction. To further validate these findings, we recommend conducting the following control experiment: exposing the hot-filtered filtrate to fresh ZSM-5 and monitoring any changes in reaction activity. This approach would help distinguish between genuine heterogeneous catalysis and potential contributions from soluble active species.

Supplementary Fig. 7 | Time-dependent conversion and hot filtration test of the reaction.

We sincerely appreciate the reviewer's valuable suggestion. We fully agree with the reviewer that the exposing the hot-filtered filtrate to fresh ZSM-5, to distinguish between genuine heterogeneous catalysis and potential contributions from soluble

active species (Fig. S7). The experimental results revealed that after adding ZSM-5 to the filtrate, no significant catalytic effect was observed, indicating that the reaction is heterogeneous and not caused by soluble active species. The slight decrease in performance may be attributed to the adsorption effect of ZSM-5.

2. ¹³C NMR characterization of the reaction products is necessary to unambiguously confirm methanol formation.

We appreciate the reviewer's insightful question regarding the ¹³C NMR, according to the reviewer's suggestions, we supplemented the ¹³C NMR data of the product (Fig. S3B). Obvious methanol-related peaks were observed, further confirming the high selectivity towards methanol.

Supplementary Fig. 3B | ¹³C NMR spectra of the products.

3. The authors achieved remarkably high H₂O₂ utilization efficiency and methanol selectivity through precise control of metal spatial arrangement and extensive comparative experiments - a highly interesting finding. To further elucidate the structural effects, it would be valuable to examine whether physical mixing of Fe/ZSM-C and Cu/ZSM-I catalysts could produce similar promotional effects. Such verification would provide more compelling evidence for the proposed structure-function relationship."

We thank the reviewer for the suggestion, which has helped to further improve the quality of our manuscript. While maintaining the total catalyst amount constant, we supplemented the experimental data for the physical mixture of Fe/ZSM-C and Cu/ZSM-I (). The results showed a methanol yield of only 3.84 mmol g_{cat}⁻¹ h⁻¹ with a

selectivity of 46.8%, which is significantly lower than that of FeCu/ZSM-CI (Fig. 1A). This further demonstrates the important role of the synergistic effect between Fe and Cu in enhancing both methanol yield and selectivity.

Fig. 1 | Catalytic performances for direct oxidation of methane. (A) Liquid product yields and methanol selectivity on different catalysts. (B) Arrhenius plots for CH₄ oxidation over FeCu/ZSM-CI, FeCu/ZSM-CC, FeCu/ZSM-II, and FeCu/ZSM-IC. (C) Cyclic experiment. Reaction Condition: 5 mg catalysts dispersed in 20 mL of 0.1 mol/L H₂O₂ aqueous solution, 80 °C and 3.5 MPa CH₄. (D-E) Comparisons with the representative catalytic performances for methanol yield and selectivity. Numbers in square brackets correspond to the entry numbers in Table S3.

4. Regarding the reaction mechanism, the authors have convincingly demonstrated

through CV measurements the *in situ* formation of high-valent metal species during catalysis - an important finding. To further strengthen the experimental validation, it would be essential to include control CV tests of the bare ZSM-5 support. This comparative analysis would unambiguously confirm that the observed redox peaks originate from the metal active centers rather than the support material itself.

Supplementary Fig. 29 | *In-situ* CV tests of catalysts for H₂O₂ activation.

We appreciate the reviewer for this very valuable comment. The *in-situ* CV profile of ZSM-5 was additionally obtained (Fig. S29C). In comparison with the CV curves of Fe/ZSM-5 and Cu/ZSM-5, no significant peaks were detected within the potential range of 0.5-0.7 V, suggesting the occurrence of dynamic valence state transitions in iron species during catalysis.

5. The authors are suggested to improve the methodological description of the radical trapping experiments (particularly the details for •CH₃, •O₂⁻, and •OH trapping), which would significantly enhance the reproducibility and reference value of the study.

Fig. 3 | Activation of methane and the Evolution of H₂O₂ into ROS. (A) EPR spectra of FeCu/ZSM-CI. (B) EPR spectra of FeCu/ZSM-CI under different reaction conditions.

We thank the reviewer for this comment. Accordingly, we have revised the methodological description of the radical trapping experiments in the manuscript.

Electron paramagnetic resonance (EPR) spectrometer was used to characterize radicals in the reaction. EPR spectra were measured on CIQTEK EPR200M with continuous-wave X band frequency, 5,5-dimethyl-1-pyrroline-N-oxide (DMPO) as the radical trap. The samples were dispersed in water-dissolved CH₄ and H₂O₂ to detect •CH₃ and •OH, and also in methanol-dissolved H₂O₂ to detect •O₂⁻.

Reviewer #3 (Remarks to the Author):

This work synthesized spatially separated FeCu/ZSM-CI catalysts, with Fe located in the inner channels and Cu on the external surface, enabling efficient direct methane to methanol by using H₂O₂ as the oxidizing agent. Interestingly, it was proposed that the tandem conversion mechanism towards methanol production was responsible for high methanol selectivity. The mechanism was further confirmed by the comparison with Fe-, Cu-, and FeCu-contact-ZSM catalysts. The paper might be publishable in Nature Communications, but the authors must address the following issues first:

1. The water is of vital importance for methane activation and methanol production using hydrogen peroxide as the oxidizing agent. Not only could it accelerate the methane activation and H₂O₂ decomposition as a co-catalyst, but also the hydrophobic effect of the channel could significantly affect the distribution of methane and hydrogen peroxide, affecting methanol selectivity. The implicit model was used in the calculations for the solvation effect for this manuscript, which could not reflect the abovementioned key roles of water. The details should be included.

We thank the reviewer for their constructive comments. In the updated computational process, we explicitly incorporated water molecules and considered the role of water in the activation of H₂O₂ and the desorption of methanol. We recalculated key steps, such as H₂O₂ decomposition, C-H bond activation, and CH₃OOH conversion, and the results show that the presence of water lowers the energy barrier for H₂O₂ decomposition into •OH on copper sites and promotes the desorption of CH₃OH from copper sites, thereby preventing further over-oxidation of methanol (Fig. 5A, 5B, and Supplementary Figs. S36-S38).

We believe these additional calculations and discussions significantly improve the accuracy and depth of the description regarding the role of water in the manuscript and provide a more comprehensive response to the key issues you raised. Thank you for helping us enhance the scientific rigor of this study.

Fig. 5 | Theoretical research on the selective production of methanol from methane. (A) Energy profiles for the conversion of CH_4 to CH_3OOH over Fe sites. (B) Energy profiles for the conversion of CH_3OOH to CH_3OH over Cu sites. (C) Diagram of the process of low temperature and high selectivity conversion of methane to methanol.

Supplementary Fig. 36 | Comparison of free energy barriers for CH_4 .

Supplementary Fig. 37 | The change of free energy of H_2O_2 to *OOH .

Supplementary Fig. 38 | The influence of H₂O on the free energy of decomposition of H₂O₂ into *OH.

2. The H₂O₂ self-decomposition is one key problem when using H₂O₂ as an oxidizing agent. The lower H-O bonding of H₂O₂ results in the preference of H₂O₂ activation and self-decomposition generally results in the lowered methane conversion (Chin. J. Catal. 51(2023), 135; Ind. Eng. Chem. Res. 63(2024), 11384). The authors need to offer more data about H₂O₂ self-decomposition.

We sincerely appreciate you raising this critical question. We fully agree with the key point rightly emphasized in the cited literature, the self-decomposition of hydrogen peroxide is a major factor limiting its efficiency as an oxidant ($\text{H}_2\text{O}_2 \rightarrow \text{H}_2\text{O} + 1/2 \text{O}_2$).

To directly address this concern, we systematically evaluated the O₂ evolution behavior from H₂O₂ decomposition in the presence of different catalysts. O₂ was detected using a gas chromatograph equipped with a TCD detector. The relevant data have been added as a new figure (Fig. S5) in the Supporting Information. The key findings are summarized as follows:

Supplementary Fig. 5 | The amount of O₂ produced by the H₂O₂ self-decomposition.

Among the various catalysts tested, H₂O₂ indeed underwent self-decomposition. The highest O₂ yield (≈ 0.28 mmol), was observed over the Fe/ZSM-C catalyst, whereas the Cu/ZSM-I catalyst produced only 0.08 mmol of O₂, indicating its lower catalytic activity toward H₂O₂ decomposition. Notably, the FeCu/ZSM-CI catalyst yielded 0.32 mmol of O₂. Compared to Fe/ZSM-C, this suggests that the presence of Cu helps stabilize H₂O₂, reducing O₂ evolution and thereby enhancing H₂O₂ utilization efficiency.

This is directly evidenced in our catalytic tests, where the FeCu/ZSM-CI catalyst achieved a remarkably high H₂O₂ utilization efficiency of 74.6%, significantly surpassing values typically reported in the literature (often below 30%). This strongly indicates that on our FeCu/ZSM-CI catalyst, H₂O₂ are preferentially activated in a controlled manner, generating valuable reactive oxygen species (such as Fe=O and •OOH) for selective methane oxidation, rather than undergoing non-selective homolytic cleavage leading to unproductive self-decomposition.

In summary, our FeCu/ZSM-CI catalyst with a spatially graded design does not exacerbate undesirable H₂O₂ consumption. On the contrary, through spatial synergy between the Fe and Cu sites, it steers H₂O₂ decomposition into an efficient, controlled, and highly selective pathway. This maximizes H₂O₂ utilization and ultimately enables high methane conversion and methanol selectivity.

3. Direct methane to methanol (DMTM) generally faces the conversion-selectivity trade-off (ACS Catal. 8(2018), 6894). The high selectivity could be easily obtained by the lower conversion that is generally lower than 20%, even though the methane

conversion is not prior to methanol deep oxidation (*Angew. Chem. Int. Ed.* 56(2017), 16464). The conversion data should be provided.

We sincerely thank the reviewer for raising this critical issue regarding the inherent challenges in DMTM. We fully agree with the reviewer and the cited literature that the conversion-selectivity trade-off is a well-known bottleneck in this field, where high methanol selectivity is typically achieved only at very low methane conversion levels to avoid over-oxidation.

Supplementary Fig. 4 | CH₄ conversion of different catalysts. Reaction Condition: 5 mg catalysts dispersed in 20 mL of 0.1 mol/L H₂O₂ aqueous solution, 80 °C and 3.5 MPa CH₄.

As requested, the methane conversion data for our FeCu/ZSM-CI catalyst under the specified reaction conditions is 1.19%. We have now explicitly included this value in the revised manuscript (Fig. S4). We acknowledge that this conversion is relatively low in absolute terms. The key advancement of our work lies in achieving an exceptionally high methanol selectivity (90.1%) at a conversion level that, while still low, results in an overall methanol yield superior to many state-of-the-art systems. To illustrate this point, we have added a comparative analysis (Table S2).

Table S2. Comparison of methane conversion rates.

Catalyst	Conversion of CH ₄ (%)	Reference
FeCu/ZSM-CI	1.19	This work
Pd ₁ -ZSM-5	0.02	Angew. Chem. Int. Ed. 2024,64, e202315343
PdCu/Z-5	0.17	Angew. Chem. Int. Ed. 2022, 61, e202204116
Au/MOR	0.25	J. Am. Chem. Soc. 2023, 145, 12928–12934
CeO ₂ @PdO@FeOx	0.02	J. Am. Chem. Soc. 2024, 146, 25870–25877
BiVO ₄ @Au	0.018	Angew. Chem. Int. Ed. 2024, 64, e202419282
Ru/ZnO	0.013	Nature Sustain. 2024,7,1171-1181
MoS ₂	0.067	Nature Catal. 2023, 6, 1052-1061
Au-ZSM-5	0.22	Nature Catal. 2022, 5, 45-54
Zn-O-Fe	0.01	Angew. Chem. Int. Ed. 2025, 64, e202510241
Cu ₉ S ₆ -Cu-C ₃ N ₄	0.024	Nature Commun. 2024, 15, 10451

This comparison demonstrates that our catalytic system delivers a higher conversion of CH₄ while maintaining high selectivity. This enhanced performance is attributed to the unique spatial gradient and synergistic Fe-Cu sites in our catalyst, which we propose selectively channel active oxygen species toward methane activation.

We concur that achieving higher conversion while retaining high selectivity remains the ultimate goal. Our current results provide a promising foundation and a novel catalyst design strategy for future optimization of reaction kinetics and process intensification.

4. How were the models for DFT simulations established? Why were T8 and T12 sites chosen as the acid sites? Why were the iron mononuclear sites coordinated with six oxygen and Cu binuclear sites chosen as the models? In particular, small Cu particles or clusters would be located on the external surface when following the experimental observation in the manuscript.

We thank the reviewer for their thorough review and valuable comments regarding the theoretical calculations in our work. The questions raised about the basis for establishing the DFT models are crucial, as they directly impact how accurately the computational results reflect the experimental system. We fully agree that model

selection must be closely aligned with experimental characterization. Below, we provide a detailed explanation of the experimental and theoretical rationale behind each key decision in our modeling.

1. Overall Modeling Approach and Selection of Zeolite Acid Sites (T8, T12)

Our DFT calculations employed periodic models based on the crystal structure of ZSM-5 zeolite. The MFI framework of ZSM-5 contains 12 distinct T (Si/Al) sites.

Basis for Selection: The choice of T8 and T12 sites for constructing Brønsted acid sites (i.e., substituting Si with Al atoms and introducing H⁺ to maintain charge neutrality) is supported by extensive literature. Studies indicate that the T12 site, located at the intersection of the straight and sinusoidal channels in ZSM-5, is favorable for Al substitution, while the T8 site is conducive to metal anchoring (References: Chem. Sci., 2012, 3, 2932-2940; Catal. Lett., 1991, 11, 209-217; J. Phys. Chem. C, 2021, 125, 37, 20373-20379).

2. Coordination Environment of Fe Site

The reviewer's observation regarding the coordination environment of the iron sites is very insightful. Our selection of a mononuclear, hexacoordinated (approximately octahedral) Fe³⁺ model is primarily based on the following experimental characterization results:

XAFS Analysis: As shown in Figure S17 and Table S6 in the main text, fitting of the Fe K-edge EXAFS data reveals no Fe-Fe or Fe-O-Fe scattering paths, which first excludes the presence of Fe nanoparticles or binuclear oxo-bridged clusters. The fitting identified two main coordination shells: ~1.6 Å (Fe-O, coordination number CN = 3.1) and ~2.6 Å (Fe-O, CN = 2.8), yielding a total coordination number of approximately 6. The shorter Fe-O bond (~1.6 Å) is attributed to Fe-OH or strong interaction with the zeolite framework, while the longer bond (~2.6 Å) is assigned to coordination with framework aluminum or more distant oxygen atoms (e.g., Fe-O-Al). This coordination environment collectively points to an extra-framework mononuclear iron species located within the zeolite channels, coordinated to the framework through multiple oxygen atoms, forming a structure with relatively high coordination saturation. Considering the Fe valence state is close to +3, the model established for Fe is

3. Structure of Copper Site

Regarding the structure of the copper sites, we agree with the reviewer's point. The presence of even small Cu clusters (e.g., Cu₂, Cu₃) would almost certainly be detected by EXAFS via Cu-Cu coordination. Consistent with the AC-HAADF-STEM observations indicating the presence of small Cu clusters on the zeolite surface, we find it more reasonable to model such species. Consequently, we have revised the model and selected a [Cu₃O]²⁺ cluster as the active site. All calculations related to the Cu site have been re-performed using this updated model.

5. The reaction mechanism should be further checked. It is unreasonable for the energy profiles in Fig. 5, starting with the transition states of TS0 and TS1. Moreover, the side reaction including H₂O₂ decomposition and deep oxidation of CH₃OH and CH₃OOH should be included to confirm the proposed mechanism towards high selectivity of methanol.

We sincerely thank the reviewer for their insightful comments and valuable suggestions regarding the reaction mechanism section. The point raised about the starting point of the energy profiles is indeed crucial, and we fully agree, we have accordingly revised the figures. Moreover, the suggestion to include side reactions to validate the high methanol selectivity mechanism has been extremely valuable. We have performed additional calculations and discussions in response, which strongly support our proposed gradient activation mechanism as key to achieving high methanol selectivity.

In accordance with the reviewer's suggestions, we have systematically revised the reaction pathway diagrams and supplemented them with calculations on key competing pathways to verify the mechanism. The specific revisions are as follows:

1. Revision of Reaction Pathways and Energy Barrier Analysis

We have re-constructed the complete reaction pathways for both Fe and Cu sites. The revised energy diagrams are shown in Fig. 5A, 5B, and Supplementary Figures S36-S38.

Fe Site Pathway (Fig. 5A): This pathway now starts with the adsorption of H₂O₂

at the Fe site, followed by heterolytic cleavage leading to the formation of high-valent Fe=O and •OOH species, which subsequently activate the C-H bond in CH₄. A comparison of the activation barriers for methane activation at Fe and Cu sites shows that the Fe site exhibits a lower activation barrier (1.11 eV vs. 1.25 eV), indicating its superior capability for initial methane activation (Fig. S36).

Cu Site Pathway (Fig. 5B): This pathway begins with the adsorption of the intermediate CH₃OOH at the Cu site, followed by O-O bond cleavage to form •OCH₃, ultimately yielding CH₃OH. This clearly illustrates the critical role of the Cu site in the selective conversion of CH₃OOH to CH₃OH.

2. Competitive Analysis of H₂O₂ Decomposition Pathways

To investigate the preference of H₂O₂ decomposition at different sites, we compared the pathways leading to •OOH and •OH formation (Fig. S37-38). The results indicate that formation of •OOH is thermodynamically more favorable at the Fe site, while it is unfavorable at the Cu site. In contrast, the energy barrier for •OH formation at the Cu site is only 0.37 eV, making this pathway significantly easier. These computational results are consistent with our EPR and radical trapping experiments (Fig. 3F-H), reasonably explaining the distinct roles of Fe and Cu sites in H₂O₂ activation: Fe sites predominantly generate •OOH, whereas Cu sites favor the formation of •OH. Meanwhile, the introduction of H₂O facilitates the activation of H₂O₂.

3. Suppression Mechanism of CH₃OH Over-Oxidation

We further evaluated the possibility of deep oxidation of methanol and its reaction intermediates:

The oxidation of *CH₃O by •OH to form formaldehyde has an energy barrier of 1.33 eV, which is higher than the barrier for its conversion to CH₃OH (1.03 eV) (Fig. 5B). This indicates that, *CH₃O is less likely to undergo further oxidation to formaldehyde. Furthermore, CH₃OH exhibits weak adsorption at the Cu site. The presence of water molecules further lowers the desorption energy barrier of methanol from 0.43 eV to 0.22 eV, significantly promoting its desorption from the active site into the bulk solution. This effect effectively prevents the retention and subsequent over-oxidation of methanol at the Cu site, thereby ensuring high methanol selectivity at the

mechanistic level.

In summary, by refining the depiction of the reaction pathways and incorporating energy analyses of competing reactions, we have further confirmed that the spatial and functional synergy between Fe and Cu sites is key to achieving efficient methane activation and high methanol selectivity.

A new discussion paragraph has been added to the DFT calculation section in the main text, and the relevant figures and discussions have been incorporated into the revised manuscript.

Fig. 5 | Theoretical research on the selective production of methanol from methane. (A) Energy profiles for the conversion of CH_4 to CH_3OOH over Fe sites. (B) Energy profiles for the conversion of CH_3OOH to CH_3OH over Cu sites. (C) Diagram of the process of low temperature and high selectivity conversion of methane to methanol.

Supplementary Fig. 36 | Comparison of free energy barriers for CH_4 .

Supplementary Fig. 37 | The change of free energy of H_2O_2 to *OOH .

Supplementary Fig. 38 | The influence of H₂O on the free energy of decomposition of H₂O₂ into *OH.

6. It was suggested that iron species are in the direct channel for Fe/ZSM-C and FeCu/ZSM-CI. However, it is difficult to find the bright spots for iron species in the direct channel in Figs. 2a and 2c.

Fig. 2 | Catalyst model. (A-C) iDPC-STEM images of Fe/ZSM-C, Cu/ZSM-I, and FeCu/ZSM-CI. (D-F) Models of Fe/ZSM-C, Cu/ZSM-I, and FeCu/ZSM-CI catalysts.

Thank you for raising this insightful and professional observation. You are correct in pointing out that in conventional iDPC-STEM images, directly observing individual Fe atoms within the channels is indeed highly challenging due to the minimal contrast difference between the ZSM-5 framework (primarily composed of Si, Al, and O) and

the isolated, dispersed single Fe atoms, especially in thicker sample regions. Nevertheless, we have observed metal-related bright spots within the ZSM-5 channels in our iDPC-STEM images (Fig. 2A). Moreover, our conclusion that "Fe species are located within the straight channels" is not solely based on direct iDPC-STEM observations, but is supported by a chain of evidence from a series of complementary characterization techniques:

Semi-quantitative SEM-EDS mapping analysis: Our data show that the Fe content on the catalyst surface (0.08 wt%) is significantly lower than the bulk Fe content (0.67 wt%, as determined by ICP-AES) (Table S5). This notable "bulk > surface" concentration difference is the most direct evidence that Fe species are successfully encapsulated within the internal channels of the zeolite, rather than distributed on the external surface.

Supplementary Table S5 | The ICP-AES and SEM-EDS results for different catalysts.

Catalyst	Fe Content (wt%)	Cu Content (wt%)
FeCu/ZSM-CI (Iron acetylacetonate)	0.67	0.50
FeCu/ZSM-CI (Ferric chloride)	0.22	0.42
Fe/ZSM-C	0.66	0
Cu/ZSM-I	0	0.51
FeCu/ZSM-CI (After the reaction)	0.66	0.49
FeCu/ZSM-CI (Iron acetylacetonate)	0.08 (SEM-EDS-mapping)	0.49 (SEM-EDS-mapping)

XAFS spectroscopy analysis: In the extended X-ray absorption fine structure (EXAFS) spectra (Fig. S16-17), no Fe-Fe or Fe-O-Fe scattering paths were detected, confirming

that Fe exists as single atoms and not as nanoparticles or clusters. Furthermore, the fitted Fe-O-Al coordination path indicates that the single Fe atoms are anchored to the zeolite framework, typically at sites within the channels.

Supplementary Fig. 16 | (A) XANES spectra at Fe K-edge of FeCu/ZSM-Cl in comparison with Fe foil, FeO and Fe₂O₃. (B) Fourier transform (FT) k^3 -weighted EXAFS spectra of FeCu/ZSM-Cl in comparison with Fe foil, FeO and Fe₂O₃. (C) XANES spectra at Cu K-edge of FeCu/ZSM-Cl in comparison with Cu foil, Cu₂O and CuO. (D) Fourier transform (FT) k^3 -weighted EXAFS spectra of FeCu/ZSM-Cl in comparison with Cu foil, Cu₂O and CuO.

Supplementary Fig. 17 | EXAFS fitting analysis of FeCu/ZSM-Cl. (a-b) Fe/Cu K-edge EXAFS; (c-d) Fe/Cu K-edge EXAFS (points) and curvefit (line) for FeCu/ZSM-Cl, shown in k^3 R-space (FT magnitude and imaginary component). The dates are k^3 -weighted and not phase-corrected. (e-f) Fe/Cu K-edge EXAFS (points) and the curvefit (line) for FeCu/ZSM-Cl, shown in k^3 -weighted k-space.

Indirect evidence from catalytic performance: There is a significant difference in catalytic performance between Fe/ZSM-C (Fe inside) and Fe/ZSM-I (Fe outside), with methanol selectivity values of 37.1% and 18.4%, respectively (Fig. 1A). This strongly suggests that the location of the Fe species (inside the channels vs. on the external surface) plays a decisive role in their catalytic behavior, indirectly supporting our

successful regulation of Fe species positioning.

Fig. 1 | (A) Liquid product yields and methanol selectivity on different catalysts.

In summary, although directly imaging individual Fe atoms within the channels presents technical difficulties, the multiple characterization methods described above form a comprehensive body of evidence that robustly supports the conclusion that "Fe species are predominantly located within the internal channels of ZSM-5." We have revised the corresponding statements in the manuscript to more accurately reflect the limitations of iDPC-STEM in characterizing isolated light metal atoms and to place greater emphasis on the integrated evidence.

Once again, we sincerely appreciate your feedback, which has helped enhance the scientific rigor of our paper.

7. In addition, FeCu/ZSM-CI catalysts with Fe located in the inner channels and Cu on the external surface was characterized after the catalyst synthesis. However, the metallic center is easily to be detached from the zeolite framework during the reaction, leading to the lowered performance. The longer period, the stability test, and corresponding catalyst characterization should be provided to confirm the active sites and reaction mechanism.

Thank you for raising this very important point. We fully agree that confirming the stability of the active sites during the reaction process and conducting long-term stability tests are crucial for verifying the authenticity of the catalyst and the proposed reaction mechanism. Following your suggestion, we have performed extended cycling experiments and conducted detailed characterization of the post-reaction catalyst. These new data have been incorporated into the revised manuscript.

Fig. 1 | (C) Cyclic experiment.

We have extended the cycling experiments to 12 cycles. As shown in the new Figure 1C, the FeCu/ZSM-CI catalyst maintained excellent stability throughout the prolonged testing, with no significant decline in methanol yield (remaining at ~ 19.8 - $20.5 \text{ mmol g}^{-1} \text{ h}^{-1}$) or selectivity (~ 89 - 91%). This result strongly demonstrates the structural stability of the catalyst during long-term reaction.

Characterization of the Post-Reaction Catalyst:

To directly investigate the state of the active sites after reaction, we performed systematic characterization on the FeCu/ZSM-CI catalyst after 12 cycles:

Supplementary Table S5 | The ICP-AES and SEM-EDS results for different catalysts.

Catalyst	Fe Content (wt%)	Cu Content (wt%)
FeCu/ZSM-CI (Iron acetylacetonate)	0.67	0.50
FeCu/ZSM-CI (Ferric chloride)	0.22	0.42
Fe/ZSM-C	0.66	0
Cu/ZSM-I	0	0.51
FeCu/ZSM-CI (After the reaction)	0.64	0.48

FeCu/ZSM-CI (Iron acetylacetonate)	0.08 (SEM-EDS-mapping)	0.49 (SEM-EDS-mapping)
------------------------	------------------------

ICP-AES: Analysis revealed negligible leaching of Fe and Cu (< 0.03 wt%) from the spent catalyst, confirming that the metal sites are firmly anchored.

Supplementary Fig. 20 | XRD patterns of the fresh catalyst and catalyst after reaction of FeCu/ZSM-CI.

XRD: The spent catalyst retained the intact ZSM-5 crystal structure.

Supplementary Fig. 23-24 | Fe 2p and Cu 2p XPS spectra of the fresh and catalyst after reaction of FeCu/ZSM-CI.

Supplementary Fig. 25-26 | R-space of Fe and Cu R-edge EXAFS for the fresh catalyst and catalyst after reaction of FeCu/ZSM-Cl.

XPS and XAFS: XPS and XAFS analyses of the spent catalyst (Supplementary Figures S23-S26) showed no substantial changes in the oxidation states or coordination environments of Fe and Cu compared to the fresh catalyst.

Supplementary Fig. 7 | Time-dependent conversion and hot filtration test of the reaction.

Hot Filtration Test: We have added a hot filtration test (Fig. S7). After removing the solid catalyst at 1 h of reaction time, the filtrate was allowed to react under the same conditions for an additional 2 h, with no additional methanol production detected. This confirms that the catalysis is heterogeneous and not dominated by leached homogeneous metal species.

In summary, through extended stability testing and detailed structural characterization of the post-reaction catalyst, we provide compelling evidence that the

Fe-Cu dual-metal sites in the FeCu/ZSM-CI catalyst exhibit outstanding structural and chemical stability during the reaction. These rules out the possibility of deactivation due to metal leaching or agglomeration, thereby confirming the reliability of the proposed active sites and reaction mechanism.

We have added these new data and corresponding discussions in the "Results and Discussion" section of the main text. Once again, we thank you for prompting us to strengthen this critical aspect of our work.

8. Due to the high cost of H₂O₂, it is unable to be applied in the realistic catalyst system using H₂O₂ as the oxidizing agent for DMTM. Is it possible to extend the current system to the DMTM using O₂?

Thank you for raising this critical question regarding practical applications and future development directions. We fully agree with your perspective that the current use of pure H₂O₂ as an oxidant poses a major economic constraint for large-scale direct methane-to-methanol (DMTM) processes. As you suggested, developing direct catalytic systems that utilize low-cost O₂ as the terminal oxidant is one of the ultimate goals in this field.

Through extensive experiments, we have observed that methane conversion is highly challenging when O₂ is used as the oxidant (Fig. S8). However, at temperatures above 200°C, methane can be oxidized to methanol and acetic acid, albeit with relatively low yields. Notably, the FeCu/ZSM-CI catalyst demonstrated a higher acetic acid yield (19 μmol g_{cat}⁻¹) and selectivity (84.6%), while Fe/ZSM-C and Cu/ZSM-I catalysts primarily favored methanol formatio. At lower temperatures (<200 °C), the reaction was almost negligible. These results indicate that, by optimizing the reaction conditions, the FeCu/ZSM-CI catalyst can still facilitate the oxidation of methane to methanol and acetic acid using O₂ as the oxidant.

Supplementary Fig. 8 |The catalytic oxidation performance of CH₄ when O₂ acts as the oxidant. Reaction conditions: 15 mg catalysts dispersed in 15 mL aqueous solution, 210 °C, 3.5 MPa CH₄, and 0.5 MPa O₂ maintain 24 h.

In summary, although the current system relies on H₂O₂, it serves as an excellent model study, providing crucial fundamental insights and clear guidance for understanding and designing the next generation of economically viable DMTM catalysts that utilize O₂. This work lays a solid theoretical foundation and offers a clear design blueprint for developing more cost-effective catalytic systems. As suggested, we have revised the manuscript to provide a more discussion of this area in lines 180-191.

Once again, we sincerely appreciate your visionary comments.

REVIEWER COMMENTS

Dear Reviewers,

We are deeply grateful for the time and effort you have dedicated to reviewing our manuscript. In light of your valuable comments, we have thoroughly revised the manuscript and supplemented relevant data and discussion accordingly. Your insightful feedback has greatly strengthened the quality of this work. Below, we provide a point-by-point response to the reviewers' comments, detailing all corresponding revisions.

Reviewer #1:

1. While the authors do emphasize the higher performance of this material compared to earlier studies it is iterative in that its is an optimization of a known material - the new configuration of the active sites reveals some similar and some different mechanistic aspects to known studies - but is not a fundamental step change in terms of discovery - apart from in terms of activity through active site optimization - particularly methane conversion.

We sincerely thank the reviewer for this thoughtful and constructive assessment. We fully agree that this work does not introduce an entirely new material class; rather, it builds upon established zeolite-supported metal active-site systems through rational design. However, we respectfully emphasize that the contribution of this study goes beyond incremental activity optimization and instead establishes a **transferable design principle** that links *active-site spatial configuration*, *reactive oxygen species (ROS) gradients*, and *reaction pathway control* in selective methane oxidation.

First, the central advance of this work lies in a conceptual shift from conventional *composition-based optimization* to **spatial-configuration engineering of active sites**. Prior studies have primarily focused on tuning metal identity, loading, or oxidation state. In contrast, we deliberately position Fe and Cu single atoms in distinct regions of the same zeolite crystal (micropore interior versus external surface), thereby creating spatially differentiated microenvironments for oxidant activation and methane conversion. This “site compartmentalization” fundamentally alters where and how key elementary steps occur, enabling simultaneous enhancement of methane conversion, methanol selectivity, and H₂O₂ utilization efficiency, an outcome not achievable

through simple increases in active-site density.

Second, our spatial Fe-Cu synergy directly addresses a long-standing intrinsic trade-off in H₂O₂-driven methane oxidation: **strong oxidizing power versus selectivity and oxidant efficiency**. In conventional systems, enhanced oxidation capability often accelerates non-productive H₂O₂ decomposition and deep over oxidation. By coupling “generation/consumption of strongly oxidizing species” with “C-H activation and product desorption” in distinct microenvironments, our catalyst configuration effectively rewires loss pathways and shifts kinetic control, leading to a pronounced performance improvement rather than a modest gain along the same conventional route.

Third, the mechanistic conclusions are not a restatement of the known fact that Fe or Cu can activate H₂O₂/CH₄. Instead, we build an evidence based chain, from radical probe/quenching experiments, EPR, in situ spectroscopy, kinetic isotope effect (KIE), and well defined control catalysts, to identify which type of site governs which elementary step, and how the spatial configuration determines the dominant reaction pathway. In other words, we reveal a rule that “active-site configuration dictates branching in the reaction network,” rather than providing a simple repetition of previously proposed mechanisms.

Finally, regarding what constitutes a “fundamental step change,” we recognize the reviewer’s point that this often refers to a completely new material or reaction paradigm. While our catalyst class indeed extends an existing material framework, we believe the discovery level advance is at the level of strategy: we provide a clear and generalizable design approach to achieve selective methane conversion under mild conditions by constructing functional domains within one zeolite host (i.e., oxidant activation versus methane activation/oxygenate protection). This represents a substantive progress in establishing a robust design principle and a structure performance mechanism linkage, and we expect this strategy to be directly informative for other selective oxidation systems.

To directly address the reviewer’s concern, we are willing to strengthen this point in the revised manuscript. Specifically, we will refine the framing in the abstract from “dual single-atom engineering” to “spatial-configuration-driven synergy and a

transferable design principle,” and we will add quantitative comparisons with representative literature in terms of “site configuration-reaction pathway-performance metrics” (e.g., CH₄ conversion, selectivity, H₂O₂ utilization), so that the non-iterative nature of our contribution relative to prior studies is more clearly and convincingly presented.

2. Can the authors further clarify their new mechanism where they propose a Fe-O-CH₄ in the mechanism figures? What experimental evidence is there for this?

We sincerely thank the reviewer for raising critical questions regarding the proposed mechanism. We would like to further clarify the Fe-O-CH₄ species depicted in our mechanism diagram. This representation is intended to describe the activation complex formed during the activation of the C-H bond in methane by high-valent iron oxo species (Fe=O or Fe-O). Specifically, it shows the interaction between the Fe-O site and CH₄, leading to the formation of a precursor or transition state that enters the H-abstraction process. Given the transient nature of this species, it is highly difficult to directly capture or structurally resolve using conventional characterization methods. Therefore, we used the Fe-O-CH₄ notation as a simplified representation of this critical step.

Although this species cannot be directly observed, multiple independent experimental and theoretical results consistently support the involvement of high-valent Fe-O species in methane C-H activation:

1. EPR Radicals detection

EPR measurements show that •CH₃ radicals are generated only when both H₂O₂ and the FeCu/ZSM-CI catalyst are present. No •CH₃ signal was detected without the catalyst or with H₂O₂ only, demonstrating that methane activation requires catalyst-mediated C-H bond cleavage rather than free-radical chemistry from H₂O₂ decomposition.

Fig. 3. | (B) EPR spectra of FeCu/ZSM-CI under different reaction conditions.

2. DMSO probe and quenching experiments

We used DMSO as a probe to differentiate between various reactive oxygen species: $\bullet\text{OH}$ can oxidize DMSO to DMSO-OH, while high-valent metals can oxidize DMSO to DMSO_2 . Both Fe/ZSM-C and FeCu/ZSM-CI convert DMSO to DMSO_2 , whereas H_2O_2 alone and Cu/ZSM-I do not, directly evidencing the formation of high-valent Fe=O species. Moreover, DMSO quenching causes a pronounced decrease in methanol yield, confirming the critical role of these species in methane activation.

Supplementary Fig. 28 | Determination of $\text{Fe}^{\text{IV}}=\text{O}$ by DMSO

Fig. 3. | (C) Quenching experiments. (D) ¹H-NMR spectrum of DMSO oxidation.

3. KIE isotope effect (KIE)

In the kinetic isotope effect (KIE) experiments, FeCu/ZSM-CI exhibited a significant $k_H/k_D = 2.32$, much higher than those of single-metal catalysts (close to 1). This suggests that C-H bond cleavage is rate-limiting in bimetallic system, consistent with an Fe-O mediated hydrogen abstraction step.

Fig. 4 | (B-D) Kinetic isotope effect experiment of CH₄ oxidation to CH₃OH over Fe/ZSM-C, Cu/ZSM-I and FeCu/ZSM-CI. (E) Kinetic isotope effect experiment of CH₃OH production over as-prepared catalysts.

4. DFT calculations

DFT calculations revealed that the C-H activation barrier at Fe sites (1.11 eV) is lower than at Cu sites (1.25 eV), supporting the dominant role of Fe-O species in forming the methane activation complex represented by Fe-O-CH.

Supplementary Fig. 37 | Comparison of free energy barriers for CH_4 .

5. In-situ CV and EPR

In-situ CV revealed characteristic redox peaks at 0.5-0.7 V for Fe/ZSM-C and FeCu/ZSM-CI upon H_2O_2 addition (Fig. 3E, S29), which were absent in the Cu/ZSM-I and ZSM-5 samples. These distinct electrochemical responses provide direct evidence for the dynamic valence state cycling of Fe species during catalysis. Meanwhile, in-situ EPR analysis revealed a much stronger $\bullet\text{CH}_3$ signal over the Fe/ZSM-C than over Cu/ZSM-I (Fig. S30), suggesting that high-valent Fe centers are the primary active sites for C-H bond activation.

Fig. 3 | (E) In-situ CV tests of catalysts for H₂O₂ activation.

Supplementary Fig. 29 | In-situ CV tests of catalysts for H₂O₂ activation.

Supplementary Fig. 30 | EPR spectra of •CH₃ under Fe/ZSM-C and Cu/ZSM-I.

6. High-field EPR (HF-EPR)

Quasi-in situ HF-EPR directly captures the reversible valence cycle $\text{Fe}^{3+} \rightarrow$ (EPR-silent) $\text{Fe}^{(\text{IV})}=\text{O} \rightarrow \text{Fe}^{3+}$ during reaction. The disappearance of the Fe^{3+} signal upon H₂O₂ exposure and its recovery after methane introduction demonstrate that high-valent $\text{Fe}^{(\text{IV})}=\text{O}$ species act as active oxygen carriers that abstract hydrogen from CH₄ and are reduced back to Fe^{3+} , fully consistent with a radical abstraction mechanism involving an Fe-O-CH₄ activation complex.

Supplementary Fig. 31 | In-situ High-field EPR of the catalyst during reaction. Reaction conditions: 463 K, 2 h, test conditions: 15 K, microwave frequency: 240 Hz, Fe/ZSM-C.

In summary, the Fe-O-CH₄ species is intended as a mechanistically meaningful representation of high-valent Fe-O involvement in methane C-H activation rather than a discrete, long-lived intermediate. Its inclusion in the mechanism is supported by converging evidence from radical detection, probe and quenching experiments, kinetic analysis, DFT calculations, and direct spectroscopic observation of Fe valence cycling. We will further clarify this interpretation in the revised mechanism figure and accompanying text to avoid any ambiguity.